# A dystonia-like movement disorder with brain and spinal neuronal defects is caused by mutation of the mouse laminin β1 subunit, *Lamb1*

Yi Bessie Liu[1], Ambika Tewari[2], Johnny Salameh[3], Elena Arystarkhova[1], Thomas G Hampton[4], Allison Brashear[5], Laurie J Ozelius[6], Kamran Khodakhah[2], Kathleen J Sweadner[1]*

[1]Department of Neurosurgery, Massachusetts General Hospital, Harvard Medical School, Boston, United States; [2]Dominick P. Purpura Department of Neuroscience, Albert Einstein College of Medicine, New York, United States; [3]Department of Neurology, University of Massachusetts Medical School, Worcester, United States; [4]Neuroscience Discovery Core, Mouse Specifics Inc., Framingham, United States; [5]Department of Neurology, Wake Forest University School of Medicine, Winston-Salem, United States; [6]Department of Neurology, Massachusetts General Hospital, Harvard Medical School, Boston, United States

**Abstract** A new mutant mouse (lamb1t) exhibits intermittent dystonic hindlimb movements and postures when awake, and hyperextension when asleep. Experiments showed co-contraction of opposing muscle groups, and indicated that symptoms depended on the interaction of brain and spinal cord. SNP mapping and exome sequencing identified the dominant causative mutation in the *Lamb1* gene. Laminins are extracellular matrix proteins, widely expressed but also known to be important in synapse structure and plasticity. In accordance, awake recording in the cerebellum detected abnormal output from a circuit of two *Lamb1*-expressing neurons, Purkinje cells and their deep cerebellar nucleus targets, during abnormal postures. We propose that dystonia-like symptoms result from lapses in descending inhibition, exposing excess activity in intrinsic spinal circuits that coordinate muscles. The mouse is a new model for testing how dysfunction in the CNS causes specific abnormal movements and postures.

*For correspondence: sweadner@helix.mgh.harvard.edu

## Introduction

Dystonia, the third-most common human movement disorder, involves 'sustained or intermittent muscle contractions causing abnormal, often repetitive movements, postures or both' (*Albanese et al., 2013*). There is strong evidence that dystonia is a circuit disorder involving various brain regions, including sensory input, premotor and motor cortex, striatum and globus pallidus, subthalamic nucleus and parts of the thalamus, cerebellum, and the tracts connecting them (*Berardelli et al., 1998*; *Breakefield et al., 2008*; *Lehéricy et al., 2013*; *Neychev et al., 2011*; *Quartarone and Hallett, 2013*; *Thompson et al., 2011*). There is also decreased inhibition and a bias toward potentiation in synaptic plasticity (*Hallett, 2011*; *Quartarone and Pisani, 2011*). However, there is little certainty about exactly how circuit and synaptic abnormalities produce the persistent overflow of motor control, often involving only certain muscle groups and the co-contraction of opposing muscles. Until recently, there has been a lack of a phenotypically penetrant genetically defined mouse model, where circuit hypotheses for mechanisms of dystonia can be

**eLife digest** Dystonia is the third most common disorder affecting movement in humans. People with dystonia periodically experience movements that they can't control. Sometimes the movements are repetitive, for example, abnormal or spasmodic blinking. At other times, two sets of muscles that work against each other become active at the same time, which causes the body to assume a strange position. The symptoms are sometimes painful and they tend to be worse in times of stress. But many individuals with dystonia are able to develop tricks to control their symptoms. For example, some find they can stop the unwanted movements by touching their face or by walking backwards.

Researchers believe that various parts of the brain fail to communicate properly in patients with dystonia. Additionally, it is thought that the connections between nerve cells called synapses become hyperactive in these individuals. However, it is not clear exactly how these abnormalities are able to circumvent the systems that usually act to suppress unnecessary movements.

Now, Liu et al. have discovered a mutation in mice that causes dystonia-like symptoms. When the mice first wake up, or when they are placed in a new environment, one or both of their hind legs become over extended. The mice walk and climb normally when they are in their usual cage and after they have been awake for longer periods. In the experiments, the mice underwent a series of tests to determine what caused these intermittent symptoms. The experiments suggested that hyperactive synapses in the spinal cord trigger the movements, but that the brain is often able to counteract them.

Genetic experiments revealed that the mice have a mutation in the *Lamb1* gene, which encodes a protein that forms a structural support in the synapse. Next, Liu et al. examined synapses in some parts of the brain of the mutant mice. During normal movements, the levels of synapse activity in these mice were similar to those observed in normal mice. However, when abnormal movements occurred in the mutant mice, their synapses produced irregular patterns of activity. Further studies of these mice should help researchers to better understand what goes wrong in human patients with dystonia.

tested in the context of abnormal movement (*Liang et al., 2014*; *Weisheit and Dauer, 2015*). The lamb1t mouse introduced here exhibits late postnatal/young adult onset of dystonia-like hindlimb movements and postures, and it has high viability, gene penetrance, and inter-individual consistency. Several aspects of its biology have parallels with dystonia, such as post-developmental onset, an ability to overcome the symptoms, and slow progression. However, the mutant mouse also has symptoms exposed by sleep and anesthesia, and these led to the demonstration that there are circuit abnormalities in the spinal cord.

The strategy was to characterize the genetic inheritance and behavior of the mouse; do diagnostic experiments to narrow down the neural substrates; map the gene's locus and identify the mutation; and check expression of mutant protein. A dominantly-inherited *Lamb1* mutation was found. Laminins are present in the extracellular matrix (ECM) surrounding neurons where they bind to synaptic proteins, and have been implicated in synaptic and neuromuscular junction structure and plasticity (*Dityatev et al., 2010*; *Wlodarczyk et al., 2011*). The mechanistic hypothesis was tested that there is altered synaptic activity in identified laminin β1-positive neurons in the CNS of the mutant mouse.

## Results

### Origin and motor behavior

The lamb1t mouse arose spontaneously in a WT C57Bl/6N mouse. It showed dominant inheritance: 140 out of 272 (51.5%) mice with one WT parent were symptomatic. Awakening or novel environment typically elicited dystonic movements. The most prominent was hyperextension of one or both hindlimbs that was clearly hyperkinetic. Movement and postural abnormalities also included widespread (extended) legs during sitting, transiently curvy tail, strong hyperextension response to

swimming, and abnormal tail suspension reflexes (*Figure 1*). Motor behavior in novel or stressful environments (unfamiliar tray; elevated rack) is shown in *Video 1*. When unstressed in the home cage, however, the mutant mice could walk normally, climb available structures, rear up while touching the side of the cage, and climb upside down on the food rack.

Dystonia in humans often has intermittent symptoms, and intermittency was a key feature of the impairment in the mice. *Video 1* also shows a mouse that straddled a beam with dystonic-looking legs while crossing it, then exhibited almost normal gait in a repeat trial. Unstressed mice ran voluntarily on wheels (*Video 2*), making neuropathy or spasticity unlikely, but during forced treadmill running dystonic symptoms emerged. We recorded gait with a DigiGait imaging system, and gait differences between WT and mutant mice are compared in the video. Gait was abnormal and limbs were often propelled to the side (*Figure 2*).

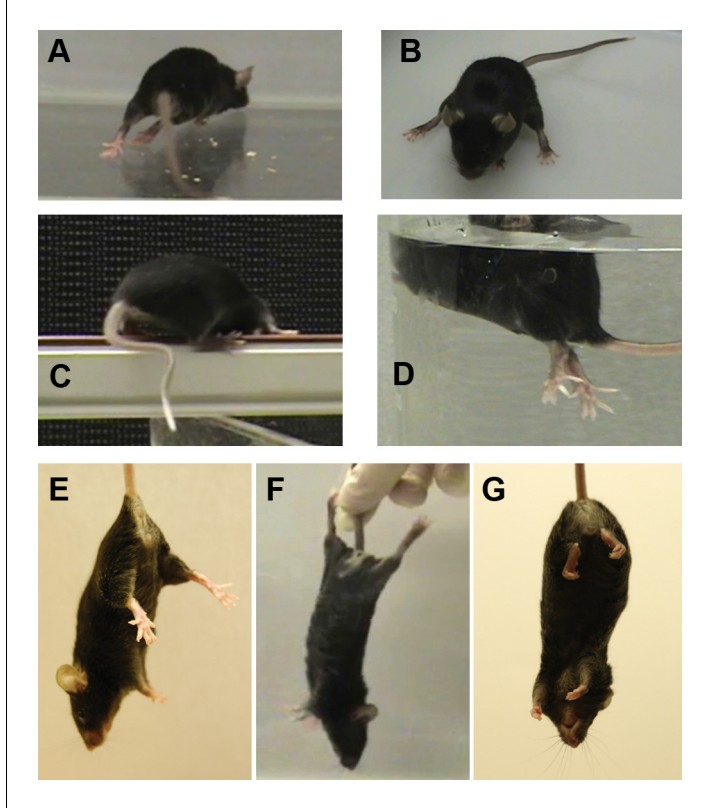

**Figure 1.** The mutant mouse had intermittent dystonic behaviors affecting hindlimbs and tail. (**A**) During ambulation, hyperextension could affect either hindlimb or both, sometimes with inversion of the foot. When hyperextension was unilateral, some mice had a preferred side et al. switched sides. Hyperextension of hindlimbs was seen at the youngest age during locomotion, but by 4 months of age was sometimes seen at rest and sometimes was bilateral. A bilateral, maximally extended posture is within a WT mouse's normal repertoire because it is shown by nursing dams straddling a large litter. (**B**) Hyperextension often continued when the animals sat. (**C**) Briefly curved tail was sometimes the first symptom in weanlings but was seen in older adults mainly when stressed. The curvature, in the plane of the floor, utilizes lateral muscle groups, and Straub tail was seldom if ever seen. (**D**) While WT mice sometimes have brief periods of rigidity and tilting when dropped in water, the mutant mice adopted an upright posture with extreme hyperextension and spread toes. They soon recovered and swam. (**E**) The normal WT reflex when suspended by the tail. (**F**) Mutants exhibited caudal hyperextensions involving one or both hindlimbs. This is also within the normal repertoire because WT exhibit a hindlimb posture like this when suspended just out of reach of an object and reaching with the forelimbs. (**G**) The mutants also exhibited transient hyperflexions of one or both hindlimbs. This was not a coordinated 'clasped' posture (limbs held together at the midline). Vibration stimulation of the knee joint in awake, hand-held mutant mice sometimes elicited strong dystonic movements when mice were released (not shown).

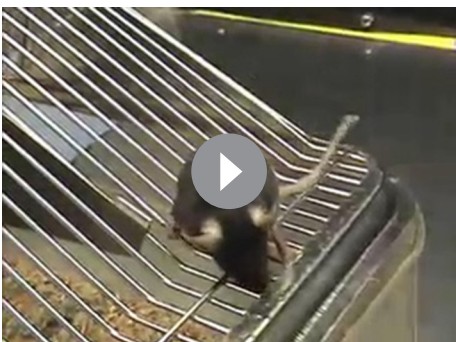

**Video 1.** *Symptoms.* Lamb1t mice displaying dystonic symptoms. 1) In a tray (novel environment) a mouse walked with kicks, adopted a wide-based sitting stance, then abruptly overcame symptoms to sit and groom. 2) A mouse with extreme hindlimb hyperextension on an elevated rack, where it also displayed curvy tail. 3) A mouse on an elevated beam traversed it with rigid hindlimbs by pulling itself across with the forelimbs. On a second trial the mouse recovered the ability to perform almost normal walking. In the home cage, mice often resumed normal motor control. Representative of many observations.

Motor symptoms were first detected between P17 and P28. Between 2 and 6 months, symptoms became more persistent, and mice often slept with hindlimbs extended. Between 6 and 12 months, there was no obvious qualitative progression separable from age, although secondary morphological changes to muscle and bone similar to those in patients may develop. Brain, spinal cord, and spinal root morphology were indistinguishable from wild type (WT), and weight gain was normal until motor symptoms become persistent.

During gentle manual assessment, hyperextended hindlimbs resisted motion at the joint. No cogwheel rigidity (as in Parkinson's Disease) or rate-dependent resistance (as in spasticity) was felt. Hindlimb stretch reflexes, elicited by slowly pulling on a foot, were present when the hindlimb was relaxed but suppressed when it was hyperextended. We detected no abnormality in the back, forelimbs, head, or axial posture, and the mice groomed and built nests well. We did not observe the following: slow paroxysmal events that wax and wane, ataxia symptoms like reeling, staggering or imbalance, tremor, loss of righting, circling, camptocormia or kyphosis, hyperekplexia, myoclonus, hopping, or any form of seizure. Social interactions appeared unexceptional, and both sexes bred successfully. The subjective impression was that the mice were alert, curious, and active.

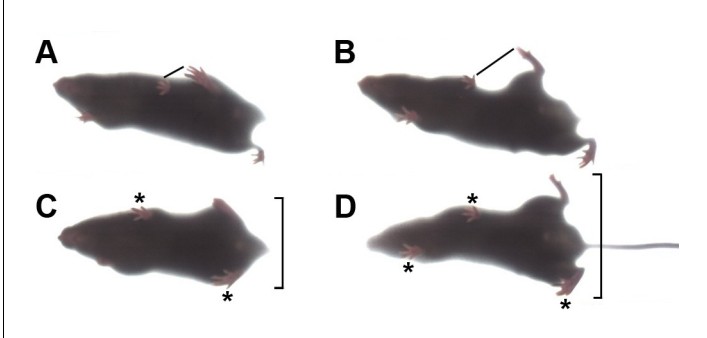

**Figure 2.** Gait abnormality in lamb1t mice. They were filmed ventrally from below the DigiGait transparent belt. The plane of focus was shallow, so feet that are seen clearly are in contact with the belt, while feet and tail that are more than a few mm above the belt are distorted or not resolved. (**A**) Treadmill-running WT mice placed their hindlimbs immediately behind the forelimbs in the alternating step pattern, and the hindlimb made contact close to the ipsilateral forelimb, as marked with a line. (**B**) In contrast, when mutant mice ran, the pattern was still alternating, but accuracy varied. In this example, the hindlimbs swung wide and did not get close to the forelimb. (**C**) Stance in mid-stride. In the WT the swinging hindlimb stayed close to the body, and mice had only two feet in contact with the belt (*). (**D**) The mutant's hindlimbs were both splayed out, and three feet were in contact with the belt. Utilizing both front feet may have compensated for deficient hindlimb mechanics. WT had three feet in contact only when breaking stride to rest briefly. The mice studied ranged in age from 63 to 132 days old, mean 109, n = 9 per group.

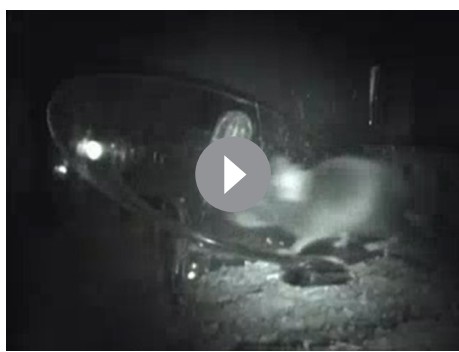

**Video 2.** *Running.* Lamb1t mice performed well when relaxed and poorly when stressed. 1) Lamb1t mice given a running wheel in the home cage ran voluntarily as soon as the lights turned off, and ran for hours as detected by a meter. This was filmed in the dark with an infrared camera. In the example shown the mouse ran smoothly but a single hindlimb hyperextension terminated the run. A magnet attached to the disk activated the meter; the viewer can use it to count rotations. Representative of n = 4. 2) Forced running on a treadmill moving at fixed speed, in contrast, was stressful and running success varied from trial to trial. 3) Slow motion ventral plane videography (DigiGait) of WT and lamb1t mice, representative of n = 9 each. WT, wild type.

## Motor skill tests

Six motor skill tests were applied. The elevated beam was used to assess motor coordination with repeated tests of the same cohort with age. The time to cross the beam was similar when WT and mutant mice were compared, but mutants exhibited many more hindlimb slips (*Figure 3A*), sometimes pulled with their forelimbs (*Video 1*), and when very affected sometimes could not stay on the beam. The rotarod tests ability to match the speed of rotation, requiring limb coordination and strength. Lamb1t mice stayed on the rod 40% as long as WT (*Figure 3B*). With male WT mice, there was also an effect of weight, illustrated by linear regression of the data (*Figure 3C*). The Olympic pool (*Figure 3D*) tested two skills combined: the time to orient when dropped in the water and the speed of swimming down a lane to a platform. Mutants were significantly slower (mutant [$6.9 \pm 0.49$ s] and WT [$2.46 \pm 0.07$ s], $p = 4.5 \times 10^{-12}$ by t-test; n = 27 WT and 28 mutant mice), but swim speed did not decline with age or weight (*Figure 3E,F*).

The activity chamber assessed spontaneous ambulation (10 min; novel environment). There was no difference in either horizontal (same beam) or ambulatory activity (different beams broken successively) between WT and mutants for either sex; that is, motor impairment did not diminish activity (two-way ANOVA, $p = 0.81$ males, $p = 0.40$ females. n = 22 WT [9 males and 13 females]; n = 20 lamb1t [10 males and 10 females]). The age range was 95–169 days, average 129 days. No difference in forelimb function was detected in the adhesive removal test (sticker applied to the forehead) (Student's two-tailed t-test, $p = 0.40$, n = 16 per group, average age 77 days). The ability to cling to a food rack from underneath utilizes skills learned in daily voluntary activity. WT mice usually confidently explored the rack. Mutants moved less and fell more (significant by t-test at all ages, 4, 6, 8, and 12 weeks, $p = 0.0018, 0.037, 4.9 \times 10^{-5}$, and $2.4 \times 10^{-12}$, respectively, and with Bonferroni correction at 4, 8, and 12 weeks; n = 14 WT, 10 males, 4 females; 7 mutants, 4 males and 3 females); however, it was not an ideal quantitative test because of variability due to the mutants adaptively using hindlimbs as hooks and wrapping their tails through the bars.

## Diagnostic tests of mechanism

We considered the possibility of peripheral neuropathy (axonal structure, sensory degeneration, or demyelinating) that would preferentially affect long axons, since symptoms predominated in the hindlimbs and tail, but no evidence for a peripheral nerve defect was found. Sciatic nerve conduction velocity (mediated by fast myelinated axons) in anesthetized mice was measured with EMG electrodes, and it was in the normal range (40 to 50 m/s, n = 3, both legs, for WT and mutant). Latency and amplitude were also like WT. Cross sections of fixed sciatic nerve (mid-thigh) showed a normal distribution of myelinated axons by toluidine blue or hematoxylin-eosin stain. Both WT and mutant had normal-appearing large and small myelinated fibers and unmyelinated fibers. We assessed the tail withdrawal reflex (mediated initially by slow unmyelinated nociceptor axons) with a tail immersion test. Time to tail flick was measured in water at 51°C. Male mice had longer average latencies to tail withdrawal than females; however, WT and mutant were not significantly different (male WT $1.99$ s $\pm$ 0.21; male mutant $2.07$ s $\pm$ 0.23; female WT $1.44 \pm 0.19$; female mutant $1.50 \pm 0.14$; mean $\pm$ SEM; male n =15 for WT and 13 for mutant; female n =9 for WT and 14 for mutant, two-way ANOVA, $p =0.0092$ for sex difference, and $p = 0.75$ for genotype effect). In the same data set, there was also

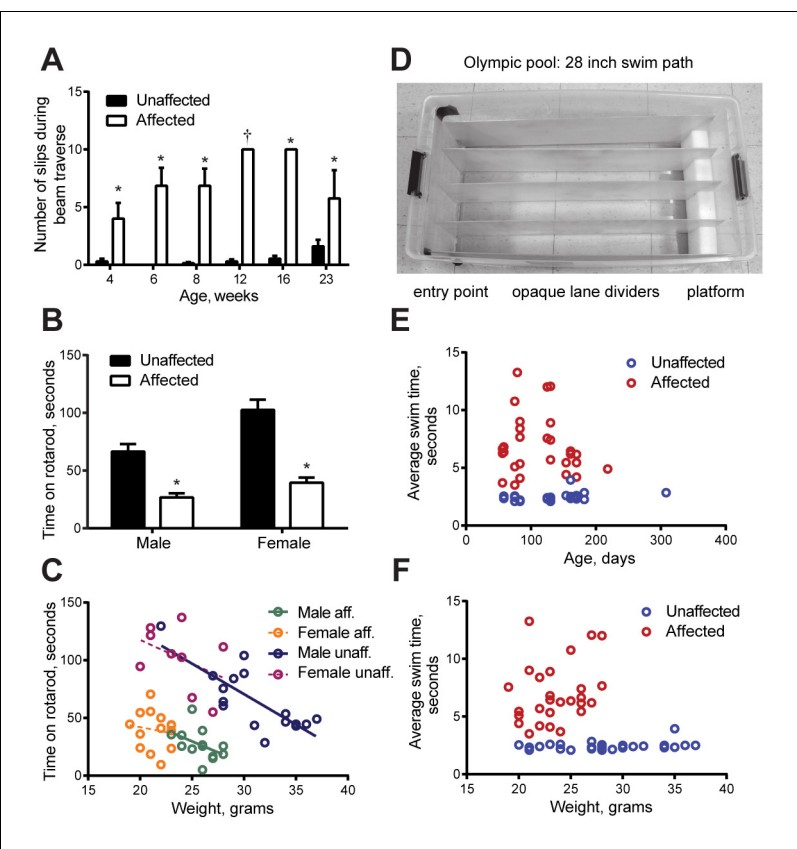

**Figure 3.** Quantitative measures of impairment. The mice (all on the C57Bl/6N background) were designated affected and unaffected before discovery of the gene. Bar graphs show mean ± SEM and two-way ANOVA was applied. (A) Crossing an elevated beam to return to the home cage was tested with one cohort repeatedly at the ages shown (n = 14 WT, 10 males, 4 females, and n = 7 lamb1t, 4 males, 3 females). Symptoms normally appeared between 3 and 4 weeks of age. Young mutant mice (1–2 months old) tended to exhibit only slips, while mature mutant mice (3–6 months old) tended to have a mix of foot slips, hyperextension, and full control, and often could not stay on the beam. Traverse time and foot slip data from trials where a mouse fell were not included in the calculations, and the mouse was given another trial. (Numbers completing the task: at 4 weeks, WT 13, mutant 7; 6 and 8 weeks, WT 14, mutant 7; 12 weeks, WT 13, mutant 1; 16 weeks, WT 13, mutant 3; 23 weeks, WT 10, mutant 4.) There was no significant difference between WT and mutant in time to cross at any age (p ranged from 0.28 to 0.76). However, the data showed a significant difference in the number of foot slips during beam traverse. At 4 weeks, p = 0.0021; 6 weeks, p = $3.7 \times 10^{-6}$; 8 weeks, p = $3.1 \times 10^{-6}$; 12 weeks, n.a.; 16 weeks, p = $3.9 \times 10^{-11}$; 23 weeks, p = 0.034 for main genotype effect (*), and Bonferroni's test confirmed significance. At 12 weeks, only one mutant mouse was able to complete the task, and the bar (†) was a single data point. (B) In the accelerating rotarod task, both male and female mutant mice showed substantially shorter latencies to falling off. After 2 days of training, two trials on the third day were averaged (WT, n = 17 male and 9 female mice; mutant, 13 males and 14 females; p = $3.4 \times 10^{-5}$ for males and $6.5 \times 10^{-7}$ for females, followed by Tukey's test). In multiple comparisons, all differences were significant except male affected vs. female affected mice. Ages ranged from 60 to 180 days (averages WT 125 days ± 47.5, SD; mutant 114 ± 44.5), and there was no trend with age. (C) Weight gain was initially normal in lamb1t mice, but they plateaued at 3–4 months, likely due to the metabolic demands of elevated muscle activity. Weight as a confounding variable is not often considered in rotarod testing. Plotting the rotarod data against weight showed it to be a continuous independent variable in WT males (linear regression for WT males had a significant slope [R square = 0.6299, F = 24.99, p = 0.0002]; slopes in the other groups tested non-significant). However, the main effect of genotype dominated the results even though male lamb1t mice weighed less. (D) The Olympic pool (empty). Mice swam down a lane to a submerged platform. (E) Swimming speed results as a function of age, and (F) as a function of weight. Each symbol is the average of two trials for one mouse. There was little overlap between genotypes, and no significant deterioration with age or weight was found by linear regression. SD, standard deviation; SEM, standard error of the mean; WT, wild type.

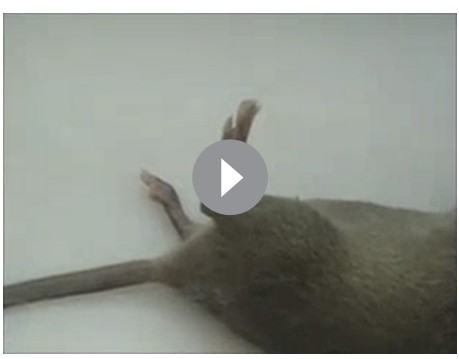

**Video 3.** *Sleep and anesthesia.* Sleep and anesthesia disinhibit abnormal spinal activity. 1) A WT female (left) and lamb1t male (right) sleeping in an igloo were filmed from underneath the cage. The mutant slept with both hindlimbs extended to different degrees. 2) Spontaneous twitching activity of another lamb1t mouse sleeping in the home cage. 3) Lamb1t mice (hybrids with different unlinked coat color gene combinations) lying anesthetized in an $O_2$/isoflurane vapor chamber. Representative of many observations.

no increase in latency with age (assessed by linear regression, range 2.4–6.6 months, average age WT 141 ± 48 days, mutant 127 ± 45 days), as would be expected for progressive neuropathy.

In sleep and in isoflurane anesthesia, twitching of lamb1t hindlimbs was often seen when mice were old enough for symptoms to be firmly established (*Video 3*). Twitching of hindlimbs during sleep was assessed at hourly intervals through the bottoms of cages with scant litter in 31 mutant mice, average 14.5 weeks old. Twitching was seen in 76 of 150 hourly observations where the mouse appeared to be asleep, indicating that it too is intermittent.

The EMG activity associated with hindlimb twitching was recorded in lamb1t mice under isoflurane anesthesia. Electrode recordings ruled out myotonia (persistent muscle firing causing prolonged contractions because of defective Cl$^-$ channels), because characteristic muscle fiber potentials (positive sharp waves and/or fibrillation potentials) with waxing and waning frequency and amplitude (*Heller et al., 1982*) were not detected, and there were periods of normal electrical silence (*Figure 4A*). The recordings also ruled out neuromyotonia (peripheral nerve hyperexcitability) and myokymia (quivering of muscles) because of the absence of either of their EMG signatures, a constant buzz of neuromuscular junction activity (*Isaacs, 1961*; *Stum et al., 2008*), or regular motor unit action potential activity in multiplet discharges (*Toyka et al., 1997*; *Zielasek et al., 2000*). We also did not detect signs of myopathy (muscle weakness), such as low-amplitude, short duration motor unit action potentials (MUAPs) (*Hanisch et al., 2014*). Plasma creatine kinase was also normal (n = 3 WT and 4 lamb1t), that is, there was no indication of muscular dystrophy.

Notably, two-electrode EMG recorded co-contraction driven by motor unit activity. Synchronized MUAPs, typically at 10 Hz, were detected with recordings from opposing (agonist and antagonist) muscles (*Figure 4B,D,F*). Synchronized polyphasic bursts very similar to those seen in sleeping blepharospasm (eyelid) and oromandibular (mouth, jaw, and tongue) dystonia patients (*Sforza et al., 1991*) correlated with twitching movements (*Figure 4C,D,E*). Most bursts were complex, with wave summation interference that obscures any underlying regularity (*Figure 4C,D*), but alternating contraction was occasionally seen (like fictive locomotion in spinalized animals) (*Figure 4E*) in addition to co-contraction. Some polyphasic bursts occurred in isolation (*Figure 4C*), others in trains of MUAPs (*Figure 4D*). Isolated large spikes that occurred randomly were not reflected in the opposing muscle (*Figure 4F*). A continuous recording from opposing muscles is in *Video 4*. To test whether co-contraction was due to a system-wide volley of excessive descending activation, electrodes were moved to the same muscles on contralateral legs or to opposing muscles on contralateral legs. No correlation of firing was observed in either case (example in *Figure 4G*).

The mouse's characteristics made it possible to ask whether the co-contraction EMG activity originated in the brain, or was intrinsic to the spinal cord. The spinal cord has local neuronal circuits (central pattern generators) required for locomotor coordination, muscle synergy, and posture. During sleep, pons and medulla normally activate inhibitors in the spinal cord, and volatile anesthetics like isoflurane activate GABA$_A$ receptors. The EMG activity exposed by sleep and anesthesia suggests two alternative hypotheses: that in lamb1t, the mutation results in the disinhibition of stimulatory signals descending from the brain, or that there is insufficient inhibition (descending from the brain or from spinal interneurons) to control overactive local spinal circuits.

Spinal transection under anesthesia was used to determine whether signals descending from the brain elicited or reduced the twitching and its underlying co-contraction. We performed this on six lamb1t mice and four WT. In all mutant mice, the activity observed under anesthesia was

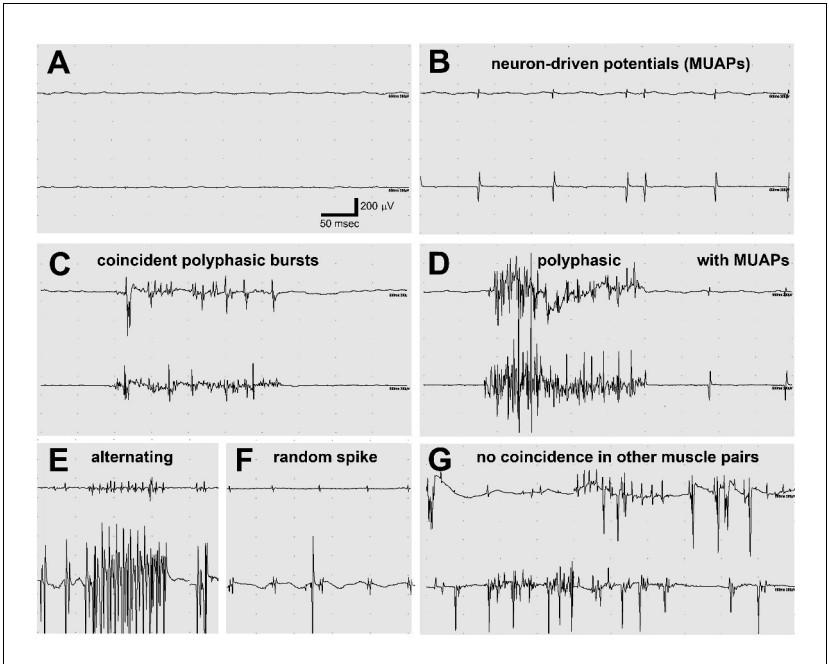

**Figure 4.** Electromyography (EMG) in the lamb1t mouse showed co-contraction. (**A–F**) Electrodes were in opposing hindlimb muscle pairs (anterior rectus femoris and posterior biceps femoris). Simultaneous activity in opposing muscles is a cardinal feature of dystonia. (**A**) Because young mice stopped showing symptoms when they warmed up after awakening, EMG was used to test for myotonia, a muscle channelopathy where a warm-up phenomenon is well-known, but results were negative. (**B,D,F**) Semi-rhythmic MUAPs (motor unit action potentials) typical of voluntary movement occurred simultaneously at 10–20 Hz under anesthesia in lamb1t mice. Sometimes recruitment of a second MUAP could be seen (as occurs with increasing force), but there was less recruitment than normal. Vigorous coincident bursts of action potentials occurred either without MUAPs (**C**), or with MUAPs (**D**). (**E**) An uncommon complex repetitive discharge at 120 Hz with antagonist muscle group alternation, a spinal discharge pattern usually associated with locomotion. (**F**) Spontaneous large single spikes were random and not seen in the opposing muscle. (**G**) Example of electrodes in the same or opposing muscles in different legs: no co-contraction or synchronization. Images are representative of n = 6 lamb1t mice. Silent recordings of littermate controls (n = 2 WT) are not shown. The mice ranged in age from 43 to 113 days, mean 59.

undiminished or slightly enhanced by spinal transection (*Video 5*). WT mice remained immobile under anesthesia before and after transection.

These observations are instructive for the mechanism of the disorder, because they rule out the possibility that the co-contraction originates in the brain. They also demonstrate aberrant intrinsic spinal activity and suggest that co-contraction is determined by the over-activation of propriospinal circuits designed to execute or oppose reflexes, motion or postural stability. The fact that the relaxed awake lamb1t mouse can suppress abnormalities and that arousal and stress produce dystonia-like symptoms shows that the brain's ability to suppress or enhance the activity of spinal cord intrinsic circuits may be a fundamental feature of dystonia.

## Identification of the gene

Locus mapping was done in hybrids between C57Bl/6N and FVB strains using SNP markers to detect recombination events. In the F1 generation, symptom onset was delayed to 5–7 weeks of age, and symptoms were less obvious. Nonetheless 10 out of 22 offspring were symptomatic. The less robust symptoms in F1 hybrids indicated that there were strain-specific modifier genes, but symptoms did not diminish with further back-crossing to FVB. For 24 individual SNP-mapped symptomatic hybrid mice only one B6 region was shared by all. The defining recombination events restricted the locus to the first 35 gigabases of chromosome 12 (*Figure 5A*).

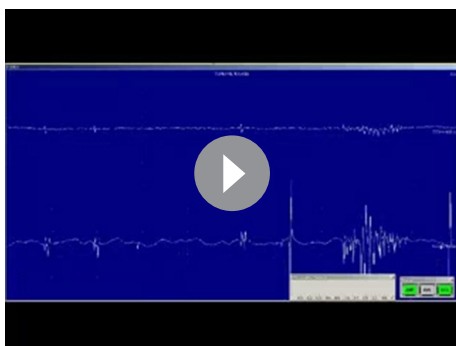

**Video 4.** *EMG.* A continuous reading of motor unit activity recorded by EMG from opposing muscles in a lamb1t mouse. Coincident timing of both MUAPs and polyphasic bursts can be seen. Representative of n = 6.

Exome sequencing was done on DNA from a WT and a mutant on the C57Bl/6NCrl background. There were nine mutant-specific variants in the chromosome 12 locus, but only one was a non-synonymous coding change. The candidate was near the closest recombination site (*Figure 5A*) and was a single base pair transversion in *Lamb1* (T5460A) that changed a leucine to a stop codon, amino acid p.Leu1730stop (*Figure 5B*). The mutation was confirmed by Sanger sequencing (*Figure 5C*) and validated by showing complete co-segregation between the mutation and the symptoms in eight littermates on the C57Bl/6 background (four with symptoms, four without), and six littermates on the mixed C57Bl/6-FVB background (three with symptoms, three without), $p < 10^{-8}$ by two-tailed t-test.

Heterozygote x heterozygote matings from the N3 hybrid generation (N3F1, N3F2, and N3F1N1F1) were used to determine whether homozygous lamb1t mice were viable. 25% of offspring of het x het crosses of FVB-B6 hybrids should have had two copies of the mutation, but instead average litter size compared to WT x het litters was reduced 28% (mean ± SEM; n = 255 mutant x WT offspring; n = 178 mutant x mutant offspring. $p = 7.9 \times 10^{-5}$, Student's two-tailed t-test). No homozygotes were found by genotyping 16 dead pups or 4 surviving runts. Intercrossing did produce some hybrids with symptoms as robust as B6 lamb1t, however, and we performed SNP mapping on 16 of those to look for homozygotes: all were heterozygous for the chromosome 12 locus. This is further evidence that homozygosity is lethal and supports the influence of unlinked modifier genes that increase or decrease symptom strength. Allele-specific PCR (*Wu et al., 2010*; *You et al., 2008*) uses otherwise-identical primers that match or mismatch the mutation at the 3' end of the primer. Allele-specific nested PCR primers were developed for routine mouse genotyping (*Figure 5D*).

## Truncated laminin β1 protein

Laminin is a trimer of three different subunits that form a cross when assembled (*Figure 6A*) (*Hohenester and Yurchenco, 2013*). The α, β, and γ subunits each have an ECM polymerization domain at the N-terminus, and α has a string of cell receptor-binding (G) domains at the C-terminus to bind signaling proteins like integrin and dystroglycan. All three subunits have C-terminal portions that let them associate with each other as a stable trimer of coiled-coil that forms an extended linear rod. The lamb1t premature stop codon deletes the last 57 amino acids from β1's coiled-coil domain. Cells have a mechanism for degrading mRNA with premature stop codons, but when a stop codon is less than 50 bases from the final exon junction, the mRNA escapes nonsense-mediated decay (*Kervestin and Jacobson, 2012*). The lamb1t mutation is in exon 32, just 36 bases upstream of the next (and final) exon junction. In immunoblots of tissue extracts of choroid plexus (a rich source of *Lamb1* mRNA; *Figure 7B*) from WT and mutant mice, laminin β1 at ~225 kDa was readily detected. The truncated form should be ~6 kDa smaller, and it was resolved when electrophoresis time was extended (*Figure 6B*). In agreement with the proposal that the truncated protein is

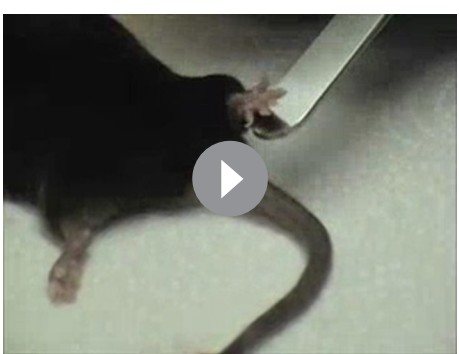

**Video 5.** *Spinal transection.* Under continuous anesthesia delivered by nose cone, two examples of hindlimb activity before and after spinal transection are shown. In the first case, rigidity and twitching increased. In the second case, gentle stimulation appeared to elicit hyperreflexia. Representative of n = 6.

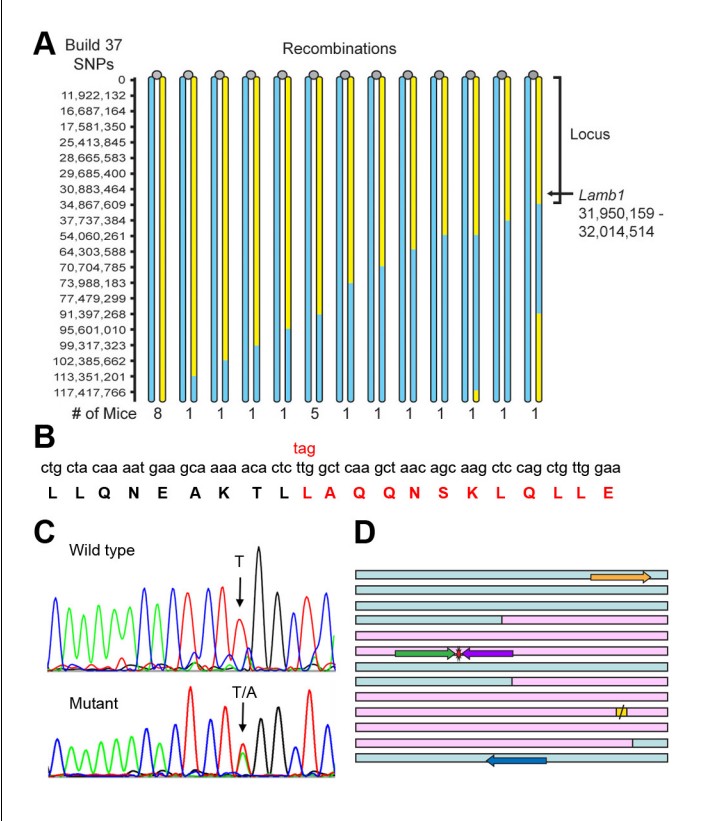

**Figure 5.** Identification of the *Lamb1* mutation. (**A**) SNP locus mapping summary for chromosome 12 in B6/FVB hybrids. The mutation is necessarily in B6 DNA (yellow); FVB DNA is blue. Mouse centromeres are at the top (gray ovals). There were no informative SNPs between base 0 and 11,922,132. (**B**) Exome sequencing result. Nucleotide and protein sequence for *Lamb1* (laminin β1) amino acids 1721 to 1741 flanking the mutation. Mutation at a single nucleotide generated a stop codon, TAG, and the sequence in red and beyond (amino acids 1730 to 1786) was truncated. Eight other variants identified by exome sequencing in the locus were in exon-flanking intron sequence or 3'UTR and not predicted to be damaging. (**C**) We validated the mutation by Sanger sequencing. The identified causative *Lamb1* mutation was not a reported variant in dbSNP, the Mouse Phenome Database, or the Sanger1 database. To date, the mutation has been verified in 33 symptomatic mice. (**D**) Allele-specific PCR design. Pink blocks are exons 32 and 33, the red symbol is the mutation, and the yellow square is the normal stop codon. The forward allele-specific primers (green) were longer than the reverse allele-specific primers (violet) because of high AT content. Each set of otherwise-identical internal primers ended with either T or A. If the mismatch is sufficiently destabilizing, priming will be absent or very low. If the mismatch is not sufficiently destabilizing, another base 5' of the mutation can be changed to reduce stability and improve selectivity; the reverse WT allele-specific primer also had a substitution of G for C at −2. Forward and reverse outside primers (gold and blue) were predicted in the flanking DNA at convenient distances from the mutation based on melting temperatures matching the allele-specific primers. Diagnostic PCR was done with the gold/violet pair with the mutation, while the gold/blue pair served as a positive control.

fully expressed, we did not observe a reduction in laminin β1 levels (*Figure 6B*); we also saw no reduction in samples from cerebellum and sciatic nerve (not shown). Whether the truncation should disrupt the coiled coil was predicted with MultiCoil. No change was calculated for the majority of coil upstream of the truncation (*Figure 6C*), implying that mutant β1 should assemble with laminin alpha and gamma subunits.

## Expression of *Lamb1* in the nervous system

*Lamb1* expression data for the mouse are available in the Allen Brain Atlas (in situ hybridization) and GENSAT (EGFP expression marker). Their data were largely in concordance where they overlapped, and confirmed published data from *Lamb1* promoter-driven β-galactosidase constructs in the mouse brain (*Sharif et al., 2004*). *Lamb1* is expressed in several sites implicated in movement disorders, including sites that showed neuropathology in rapid-onset dystonia-parkinsonism (RDP) (*Oblak et al., 2014*). These include substantia nigra compacta, cerebellar Purkinje neurons, and the

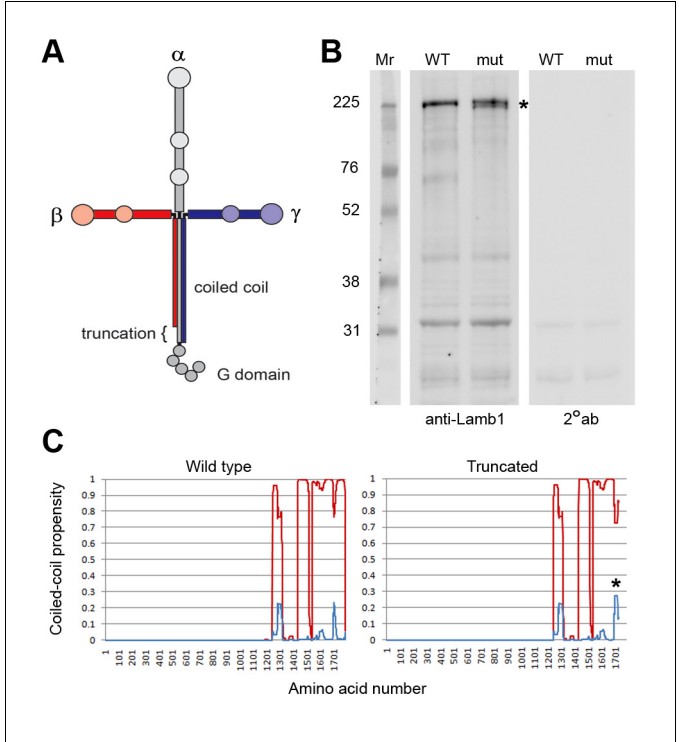

**Figure 6.** Laminin $\beta$1 protein structure. (**A**) Diagram of the laminin trimer of $\alpha$ (gray), $\beta$ (red), and $\gamma$ (blue) subunits. Circles are major globular domains and the largest ones are binding sites for each other and other extracellular matrix components. The G domain repeating globular domains are sites of attachment to cell surfaces through integrin and dystroglycan. The rod-like coiled-coil trimer domain, a quaternary structure, is altered by truncation of the last 57 amino acids in the $\beta$1 subunit in lamb1t. Coiled-coils are composed of alpha-helices tightly wound together, and the absence of a portion should destabilize the $\alpha$ and $\gamma$ segments at that site. (**B**) SDS gel electrophoresis (NuPage 4–12% polyacrylamide gradient MES gels run 50% longer than normal) followed by immunoblot with laminin $\beta$1-specific antibody. The doublet resolved in the mutant (*) is assumed to be the proteins produced from WT and mutant alleles. (**C**) We used MultiCoil software to calculate the propensity of protein sequence to form two-stranded or three-stranded coiled coils (http://groups.csail.mit.edu/cb/multicoil/cgi-bin/multicoil.cgi). Laminin $\beta$1 is strongly predicted to form three-stranded coiled coils. Just upstream of the truncation, there was a slight decrease in triple-stranded coil propensity (red) and slight increase in double-strand coil propensity (*, blue) for 27 amino acids, but no change for the rest.

deep cerebellar nuclei (DCN). *Lamb1* is also expressed in the striatum, where the low density of labeled cells indicates a discrete population of interneurons (***Figure 7A***). Only a few neurons of the cortex were labeled, requiring identification. Other cells and nuclei were also stained well, including the hippocampus and thalamus, but overall staining was light and selective. Interestingly, distributions characteristic of either astrocytes or oligodendrocytes were not seen.

The positive Purkinje neurons and DCN (***Figure 7B,C***) are the output pathway of the cerebellum. *Lamb1* seems to not be expressed in some excitatory inputs to Purkinje cells (granule cells, or the inferior olivary nucleus, ***Figure 7D***) but is expressed in inhibitory inputs from molecular layer cells that probably are basket and stellate cells (***Figure 7B,C***). In the spinal cord, there is diffuse stain in lamina 1, and strong expression in a sparse population of interneurons in lamina 2 or 3 of the dorsal horn (***Figure 7E–G***). These are laminae where unmyelinated nociceptive sensory neurons terminate (***Gardiner, 2011***). Motor neurons appear to have little or no mRNA for *Lamb1*, and the EGFP marker expressed in ventral horn was in blood vessels (***Figure 7E***).

## Awake in vivo cerebellar electrophysiology

Among the *Lamb1*-positive neurons, cerebellar Purkinje neurons and deep cerebellar nuclei (DCN) neurons have well-characterized synaptic interactions. We hypothesized that firing abnormalities

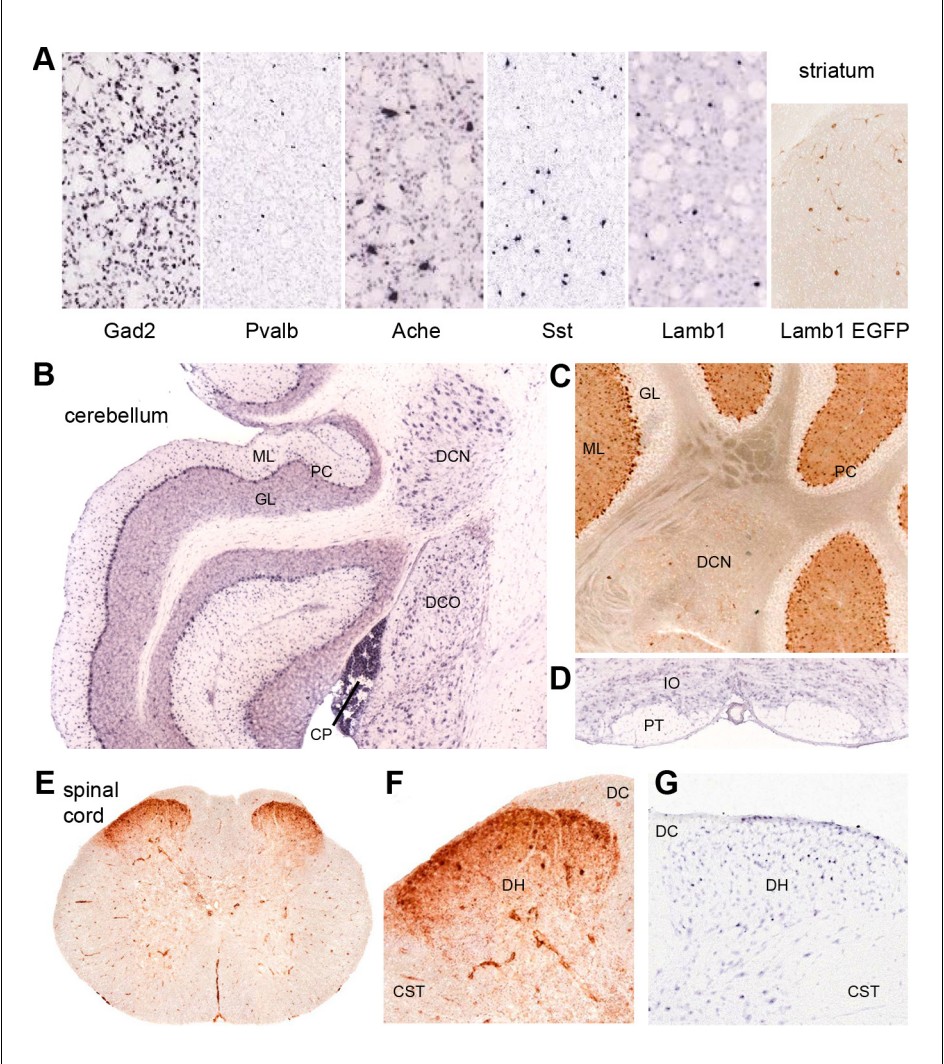

**Figure 7.** Discrete nervous system expression of *Lamb1*. Images are reproduced from the Allen Brain Atlas (Allen Institute for Brain Science) (A,B,D,G), or GENSAT (C,E,F). (A) In situ hybridization in striatum on lightly counter-stained mouse sections. *Gad2* signal marks the abundant medium spiny neurons. *Pvalb, Ache*, and *Sst* are markers of different interneuron populations. Cholinergic neurons (*Ache*) are very large. *Lamb1*-positive cells are apparently in an interneuron population, but unlikely to be cholinergic because of their small size. EGFP expression confirms the in situ hybridization signal for *Lamb1*. (B) Cerebellum. *Lamb1* in situ hybridization is high in choroid plexus (CP) and Purkinje cells (PC), and expressed at a lower level in the deep cerebellar nuclei (DCN), including all three, dentate, interpositus and fastigial. Label in dorsal cochlear nucleus (DCO) is also present. (ML) molecular layer where Purkinje cells arborize. There are strongly stained interneurons scattered in the molecular layer. (GL) granular layer, where the granule cells, the abundant excitatory inputs to Purkinje cells, do not express *Lamb1*. There are sparse labeled cells, however. (C) EGFP expression supports the findings. (D) Another major excitatory input to Purkinje cells, the climbing fibers, come from the inferior olivary nucleus (IO), which was not labeled for *Lamb1*. (PT) pyramidal tract. (E) Lumbar spinal cord. EGFP expression was in a diffuse band in lamina 1 of the spinal cord, and in strongly labeled scattered cells in lamina 2 or 3. There was little or no label in other structures other than blood vessels. (F) Higher magnification with dorsal horn (DH), dorsal sensory column (DC), and corticospinal tracts (CST) indicated. (G) Available in situ hybridization in spinal cord was faint, but a similar subpopulation of cells in lamina 2 or 3 as well as some cells at the surface were labeled. Reproduced with permission.

might be detected in those cells if laminin β1 mutation is acting at synapses. DCN provides a major output from the cerebellum to ensure coordinated motor activity, and is one part of the circuits important for dystonia. Single-unit extracellular recordings of spontaneous activity of Purkinje cells and cells in the DCN were performed in awake head-restrained mice. *Figure 8* presents the data as box and whisker plots to show differences in median, interquartile range, and skew, while *Table 1* has the calculated means and statistical differences. Recordings revealed irregularly firing neurons during abnormal postures in the lamb1t mice compared to WT (*Figure 8A,E*). To quantify the

changes in firing and analyze lamb1t firing patterns during both normal and abnormal postures, several parameters were examined. The average firing rate (defined as the number of spikes divided by the recorded time) was decreased in the DCN in the mutant during abnormal postures, whereas there was no significant difference between the WT and the mutant during normal postures (*Figure 8B*). The predominant firing rate (defined as the reciprocal of the mode interspike interval) of the mutant mice was significantly higher during both normal and abnormal postures compared to WT, but highest with abnormal postures (*Figure 8C*). To measure the irregularity of firing in all conditions, the coefficient of variation of the interspike interval (CV ISI), defined as the standard deviation of the interspike interval divided by its mean, was calculated. There was a significant increase in the CV ISI of DCN cells in the mutant during abnormal postures compared to WT or to mutants during normal postures, confirming that cells in the DCN of mutant animals fire irregularly during abnormal postures.

Since a major source of input to the DCN comes from Purkinje cells, single-unit recordings from Purkinje cells were performed on the same mice to determine whether these cells are a likely contributor to the irregularity of firing in the DCN. Similar to cells in the DCN, Purkinje cells in mutant mice appeared to fire irregularly during abnormal postures when compared to WT mice (*Figure 8E*). Like in the DCN, average rate was higher in lamb1t during normal postures, but was similar to WT,

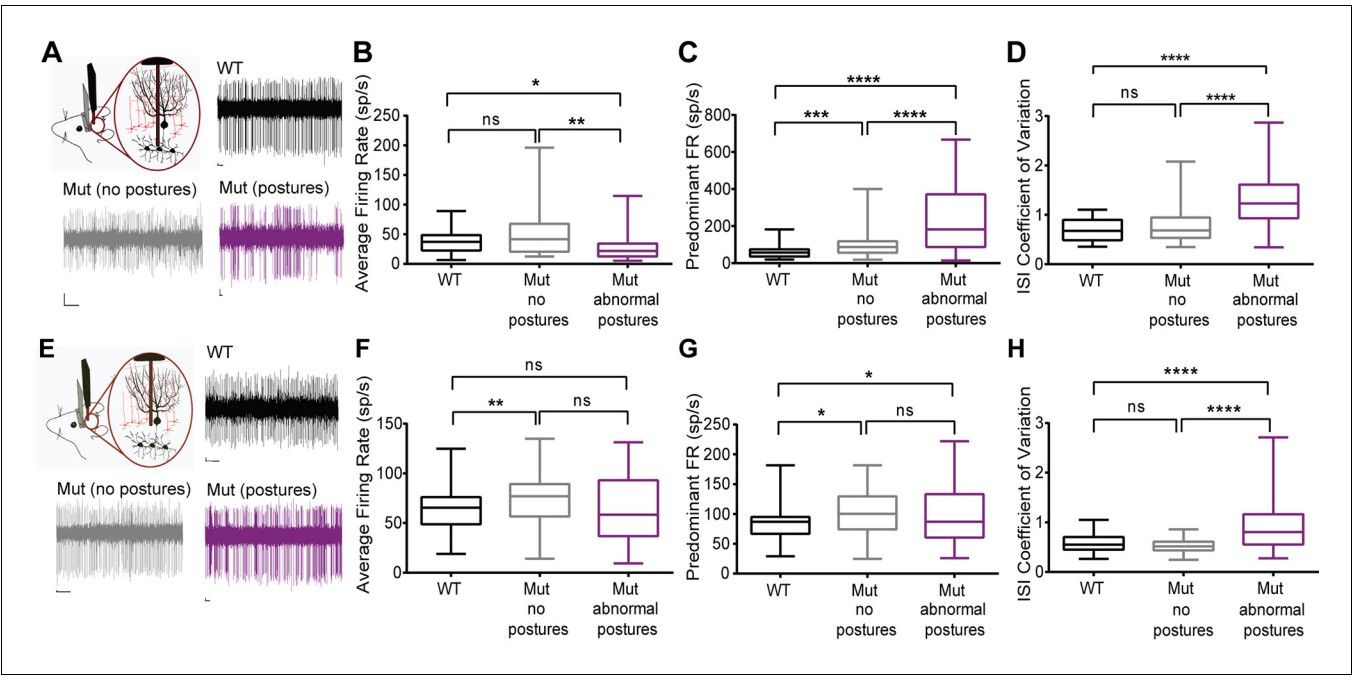

**Figure 8.** Cerebellar neurons in lamb1t mice during abnormal postures exhibited high-frequency bursts. Extracellular recordings were performed in vivoin awake head-restrained wild-type and mutant mice. (**A**) Representative raw traces of spontaneous single-unit recordings in cells of the DCN show abnormal burst firing of cells in the mutant during abnormal postures (magenta) when compared to periods where the mouse did not exhibit abnormal postures (gray) and also compared to the WT (black). (**B**) Even though there was no significant difference in the average firing rate of cells in the DCN of WT compared to the mutant without postures, during abnormal postures the average firing rate decreased significantly. (*p<0.05, **p<0.01, mean ± SEM) (**C**) The data were binned to construct a histogram of the interspike intervals (ISI) from which the peak value (the mode of the distribution) was determined for each cell. The predominant firing rate was calculated as the reciprocal of the mode of the ISI. During abnormal postures in the mutant animal, the predominant firing rate of cells in the DCN was more than 3-fold higher compared to the WT and twofold higher compared to conditions when the mutant was not displaying abnormal postures (****p<0.0001, ***p<0.001) (**D**) The coefficient of variation of the ISI (CV ISI) was significantly higher in the mutant during abnormal postures (****p<0.0001), whereas during conditions of no postures the CV ISI was similar to the WT (**E**) Raw traces showing irregular firing of Purkinje cells in the mutant when compared to the WT. (**F**) When compared to the WT, the average firing rate was statistically higher in the mutant with no postures (**p<0.01) but not significant in the mutant during abnormal postures. However, there was no significant difference between mutants with and without abnormal postures. (**G**) Predominant firing rate was significantly higher in the mutant with and without postures (*p<0.05) when compared to the WT. (**H**) Similar to cells in the DCN, the CV ISI was significantly increased in the mutant Purkinje cells during abnormal postures compared to normal postures and to WT (****p<0.0001). Scale bars in raw traces: X-axis: 300 ms and Y-axis: 20 µV. WT, wild type.

**Table 1.** Cerebellar firing patterns.

| | WT | lamb1t normal | p, norm. vs. WT | lamb1t abnormal | p, abnorm. vs. WT | p, norm vs. abnorm. |
|---|---|---|---|---|---|---|
| *Deep cerebellar nuclei neurons* | | | | | | |
| # of cells | 26 | 42 | | 56 | | |
| average f.r. | 37.4 ± 3.7 | 50.3 ± 5.9 | 0.3499 | 26.7 ± 2.9 | 0.0109 | 0.0029 |
| predominant f.r. | 61.0 ± 6.9 | 110.1 ± 13.8 | 0.0004 | 221.9 ± 20.9 | <0.0001 | 0.0001 |
| CV ISI | 0.677 ± 0.05 | 0.751 ± 0.05 | 0.6542 | 1.272 ± 0.07 | <0.0001 | <0.0001 |
| *Purkinje neurons* | | | | | | |
| # of cells | 61 | 76 | | 45 | | |
| average f.r. | 62.9 ± 2.8 | 73.6 ± 2.8 | 0.0052 | 62.7 ± 5.0 | 0.3400 | 0.0619 |
| predominant f.r. | 83.2 ± 3.4 | 98.7 ± 3.6 | 0.0141 | 103.8 ± 8.1 | 0.0128 | 0.4462 |
| CV ISI | 0.578 ± 0.02 | 0.531 ± 0.01 | 0.3036 | 0.919 ± 0.07 | <0.0001 | <0.0001 |

Means and significance of firing rates and coefficients of variation. Definitions: f.r., firing rate. CV ISI, coefficient of variation of the interspike interval; that is, variability. The data are means ± SEM, and the p values were determined by the Kolmogorov-Smirnov test for non-parametric distributions, which can be seen in **Figure 8**. N = 3 WT and N = 4 lamb1t mice; n for each class of cell recording is given in the table.

not lower, when abnormal postures were present (*Figure 8F*). The predominant firing rate in mutants was higher during both normal and abnormal postures than in WT (*Figure 8G*). Also like in the DCN, the CV ISI of mutants during normal postures was not significantly different from WT, but during abnormal postures CV ISI was much higher (*Figure 8H*).

In summary, the electrophysiological data demonstrate that Purkinje cells and cells in the DCN of the lamb1t mouse fire irregularly with high-frequency bursts during abnormal postures, and that even during normal postures the mutant exhibits an increased predominant firing rate, but without an increased coefficient of variation. This indicates that the *Lamb1* mutation results in a change in the intrinsic properties of both Purkinje cells and DCN neurons or in their synaptic inputs. The data do not exclude the involvement, or the possible primacy, of other circuits in the manifestation of symptoms, and much is yet to be learned from this mutant.

## Discussion

Laminin β1 is present in the ECM in many tissues and in different subunit isoform combinations, each laminin composed of one α, one β, and one γ subunit. Laminins are important system-wide for adhesion, migration, differentiation, and structural integrity, and they activate cell-surface integrins, syndecan, dystroglycan, and other signaling proteins via the α subunit C-terminus (*Hohenester and Yurchenco, 2013*). System-wide and developmental effects of the dominant heterozygous mutation in lamb1t have not been detected. Any effects in heterozygotes must be subtle because of the robust health and longevity of the mouse, the post-developmental onset of motor symptoms, and their intermittent nature. Here, we discuss the significance of the genetic and structural features of the mutation; potential roles of C-terminal laminin mutations in synapses; and the implications of the results for movement disorder circuitry.

### Genotype, structure, and inheritance

The 57 amino acid truncation mutation produces a dominantly inherited phenotype with ~100% penetrance. A phenotype was not exhibited by heterozygous *Lamb1* knockout mice (*Yin et al., 2003*). Since the heterozygotes of the knockouts and lamb1t were on the same strain background, the dominant phenotype of lamb1t must be due to assembly of altered protein rather than haploinsufficiency, suggesting a gain of function. Based on the analysis of coiled-coil propensity, the sequence predicts that the small truncation of β1 should assemble with α and γ subunits because the coiled-coil domain required for trimerization is almost intact (including residues for interchain disulfide bonds). Agrin, another important synapse-organizing ECM protein, binds to a central portion of

laminin's coiled-coil domain, and interacts with γ1 (*Kammerer et al., 1999*), and that interaction may be preserved. The β1 truncation is unlikely to impair laminin trimer integration into ECM, because integration utilizes the self-polymerization domains at the N-termini of the three subunits. However, like the *Lamb1* knockout (*Miner et al., 2004*), lamb1t mice were embryonic lethal when homozygous, so it is essential to have one good copy in early development.

Lamb1t's mutation is at the receptor-binding end of the laminin complex, and there are some precedents for the kinds of effects it may have. Deletion of just three amino acids from the C-terminus of either γ1 or γ2 abrogated binding of laminin to the receptor integrin, as did substitution of glutamine for glutamate at the -3 position (*Ido et al., 2007*). Chimeras of the last 20 residues of laminin β1 and β2 determined selectivity among different integrins (*Taniguchi et al., 2009*). Synaptic localization of muscle-secreted laminin β2 was determined to be controlled by a 16 amino acid motif 100 residues upstream of the C-terminus, which may still be exposed (*Martin et al., 1995*). Thus, the 57-residue truncation in lamb1t may alter association with cell-surface receptors and affect signaling, without necessarily affecting the structural properties of laminin in the ECM. Alternatively, the truncation may destabilize the coiled-coil domain of the complex enough to make it more sensitive to regulated extracellular proteases, changing the kinetics of protease-mediated synaptic processes (*Dityatev et al., 2010*; *Wlodarczyk et al., 2011*).

Two *LAMB1* mutations in humans show a genotype-structure-phenotype relationship that contrasts with lamb1t. The mutations in two consanguineous families were recessive and produced a severe developmental brain disorder when homozygous, cobblestone lissencephaly (COB), caused by defective basement lamina at the pial surface (*Radmanesh et al., 2013*). This outcome was less severe than the in utero lethality of homozygous lamb1t mutation, but similar brain dysplasia was seen in homozygous mouse knockouts of *Lamb2* and *Lamc3* (*Radner et al., 2013*). The COB mutations had much larger truncations (via frameshifts) than lamb1t's, which eliminated the coiled-coil domain and would prevent trimerization if expressed. The frameshifts should also trigger nonsense-mediated mRNA decay (*Kervestin and Jacobson, 2012*), which in homozygous patients would be equivalent to knockout of *LAMB1*. Either problem would explain the fact that COB gene carriers in the families were asymptomatic, like the asymptomatic heterozygous *Lamb1* knockout mice. Given the high level of mRNA for *Lamb1* in choroid plexus of mice, it is of interest that the *LAMB1* homozygous COB patients also had hydrocephalus, likely due to impaired ECM structural integrity in the choroid plexus. We predict that if laminin mutations functionally equivalent to lamb1t are eventually found in patients, they should be limited to residues that interfere with receptor binding, but not with laminin assembly and ECM integrity.

## Laminin's roles in synapses

ECM is a degradable structure that stabilizes neuronal circuitry, is present in the synapse, and is actively remodeled to facilitate plasticity (*Dityatev et al., 2010*; *Wlodarczyk et al., 2011*). Activity-dependent synaptic plasticity, such as long-term potentiation mediated by the expansion of spines, entails remodeling of the ECM via regulated proteases. Tissue plasminogen activator (tPA) is secreted; then plasmin cleaves laminin and activates matrix metalloprotease 9 (MMP-9); then MMP-9 digests a laminin receptor, dystroglycan (*Wlodarczyk et al., 2011*). If this is facilitated by the lamb1t mutation, it may support evidence that synaptic plasticity is increased in dystonia (*Quartarone and Pisani, 2011*)

Laminin is well-studied for its role in controlling neuromuscular junction formation and ultrastructure at least partly through the receptor-binding domain (*Martin et al., 1995*; *Nishimune et al., 2004*; *Singhal and Martin, 2011*). For example, laminins with β2 (*Lamb2*) complex with voltage-gated calcium channels to organize active zones, and signal through dystroglycan to modify neuromuscular junction structure and stability. Much is still to be learned about laminin β1 in the CNS, but synaptic vesicle protein 2 (SV2) is a presynaptic ECM receptor, and binds laminins with β1 (*Son et al., 2000*). One lab has produced mechanistic evidence for a role for laminin β1 in synaptic plasticity. Molecular mechanisms of learning and memory were investigated by gene manipulation in vivo in rat hippocampus in combination with water maze testing (*Yang et al., 2011*). Maze learning decreased laminin β1, and conversely laminin β1 overexpression impaired maze performance. Pertinent to the signaling function of laminins, laminin β1 impaired learning through activation of ERK/MAPK and SGK1 (*Yang et al., 2011*). A plasticity cascade entails JAK/STAT activation (*Nicolas et al., 2012*). Maze learning downregulated STAT1, and STAT1 overexpression impaired

maze learning while strongly upregulating *Lamb1*. In a crucial experiment, the effect of STAT1 on maze performance was blocked by siRNA for *Lamb1* (*Hsu et al., 2014*). Changes in NMDA receptor subunits were upstream of STAT1. This predicts that laminin β1 is in the middle or end of a synaptic plasticity cascade.

The evidence in this report does not prove that the C-terminal truncation in laminin β1 affects synaptic plasticity; however, the altered firing patterns detected in deep nuclei in the cerebellum provide strong cellular evidence for an effect on synapse function or organization. Very similar irregular firing was seen in both DCN and Purkinje cells in a pharmacological model of *ATP1A3* dystonia (Dyt12) (*Fremont et al., 2014*). DCN neurons also had neuropathology in rapid-onset dystonia-parkinsonism specimens (*ATP1A3*) (*Oblak et al., 2014*), and they are the source of the cerebello-thalamo-cortical tracts that show alterations in *TOR1A, THAP1,* and *ATP1A3* dystonia patients by diffusion tensor imaging (*Argyelan et al., 2009*; *Lehéricy et al., 2013*; *Whitlow et al., 2012*). In lamb1t, the irregular firing of DCN neurons could be caused by an increase in the efficacy of inhibitory Purkinje cell-to-DCN synapses (*D'Angelo, 2014*). The detected abnormalities of a synapse in lamb1t support a role for cerebellar abnormalities in dystonic symptoms (*Fremont et al., 2014*; *Wilson and Hess, 2013*). This of course does not rule out effects in other circuits with *Lamb1*-positive neurons, like striatum and of course spinal cord, that may be required for lamb1t's aberrant motor control.

## Implications for movement disorder circuitry

The lamb1t mouse is a phenotypic model with overt dystonia-like symptoms when awake. *Lamb1* gene expression is seen in selected neurons in the basal ganglia, cerebellum, spinal cord, and other locations, but it is not expressed universally in the CNS. Identification of *Lamb1*-positive neurons and their interactions will be a first step to investigating the underlying mechanisms, as begun here with cerebellar recordings.

Do the dystonia-like symptoms of lamb1t represent dystonia as understood clinically? Four theoretical frameworks coexist in human and animal dystonia research: roles for the basal ganglia, cerebellum, motor areas and the pathways that connect them; alterations in sensorimotor integration; reductions in neuronal inhibition; and increases in the potentiation side of synaptic plasticity (*Berardelli et al., 1998*; *Breakefield et al., 2008*; *Kreitzer and Malenka, 2008*; *Lehéricy et al., 2013*; *Neychev et al., 2011*; *Quartarone and Hallett, 2013*; *Quartarone and Pisani, 2011*; *Tanabe et al., 2009*; *Thompson et al., 2011*). None of these mechanistic aspects can be ruled out in the lamb1t mouse, and the possibility that the laminin β1 truncation affects synapse organization and plasticity is compatible with all. The post-development onset, slow progression, and plateau of symptoms resembles progression and stabilization in human dystonias. The ability to overcome symptoms resembles the human ability to use sensory tricks, such as touching the face, to temporarily overcome dystonia (*Ramos et al., 2014*). However, in dystonic humans, a rapid alternation of affected muscle groups is unlikely, as seen here when mice switch affected hindlimbs. Furthermore, while electromyographic (EMG) activity persists in sleep in a few forms of human dystonia (*Sforza et al., 1991*; *Stamelou et al., 2012*), it is not typical. Research on focal dystonia patients suggests altered spinal reflexes and failure of reciprocal inhibition in spinal circuits as the immediate cause of co-contraction (*Berardelli et al., 1998*; *Panizza et al., 1990*; *Sabbahi et al., 2003*; *Tanabe et al., 2009*; *Thompson et al., 2011*). Patient research also indicates a role for the brain to supply inhibitory instructions to spinal interneurons (*Berardelli et al., 1998*; *Hallett, 2011*; *Quartarone and Hallett, 2013*). These aspects are consistent with traits of the lamb1t mouse, and share with it the convergence of abnormalities in the brain and spinal cord. However, CNS lesions causing human secondary dystonia tend to be in basal ganglia and their connections (*Thompson et al., 2011*). This would be conceptually consistent with the lamb1t mouse's characteristics if typical dystonia-causing human lesions damage the ability of the brain to control (inhibit) circuits in the spinal cord.

The hindlimb locus of lamb1t symptoms may be characteristic of the species: hindlimbs have a dominant place in sensorimotor cerebellar integration in mice compared to primates (*Logan and Robertson, 1986*; *Raike et al., 2013*), and hindlimb symptoms are also prominent in other rodent non-paroxysmal dystonia models (*LeDoux, 2011*; *Tanabe et al., 2012*; *Weisheit and Dauer, 2015*). The lack of overt dystonic symptoms in mouse models with engineered dominantly inherited human dystonia mutations may be also influenced by a higher level of automaticity of the control of spinal

cord in rodents relative to primates, requiring a stronger spinal physiological abnormality to allow manifestation of symptoms more obvious than hindlimb slips.

The presence of *Lamb1*-expressing neurons in the spinal cord presumably underlies the observed abnormal spinal circuit activity. Neurons that express *Lamb1* in dorsal horn may mediate sensory input, or may regulate propriospinal circuits that produce reflexes, coordinate muscles, or stabilize posture. If DRG neurons express *Lamb1* in adults, aberrant sensory input could also have a role. Caution should be used before assuming that the causative functional defect is in the spinal cord, however, because one fundamental way that motor output is controlled is gradual enhancement of spinal reflex strength by the brain via descending signals, causing long-term scaling changes in the spinal cord (*Wang et al., 2012*). Spinal circuit abnormalities might accrue slowly secondary to a primary defect in the brain. This would be congruent with slow symptom progression in the mouse, and brain–spinal cord interaction might underlie the slow therapeutic response to deep brain stimulation in patients.

Available data ruled out other diagnoses such as myotonia, neuromyotonia, neuropathy, or muscular dystrophy. The features of lamb1t mice are *ab initio* different from spasticity (a form of hypertonicity, clinically defined as velocity-dependent resistance to muscle stretch). In spinal injury or upper motor neuron disease, spasticity results from spinal reflex changes secondary to missing input from upper motor neurons (*Sheean, 2002*). Here, there was never paralysis preceding onset of dystonic symptoms; the mice could run voluntarily; and the symptoms were intermittent. It remains to be seen, however, whether abnormalities in descending inhibition, for example in the output of the pontine inhibitory region, can elicit spasticity-like changes in the spinal cord. Changes in spinal circuits (such as the serotonin supersensitivity of coordinated muscle activation by central pattern generators in spasticity [*Husch et al., 2012*]) could be endophenotypes that are actually shared by dystonia and spasticity, globally or focally. Clinical spasticity can also be accompanied by spasms of muscle contraction, which are potentially related to the motor unit-driven twitches seen in lamb1t under sleep and anesthesia. At this stage, the relationship of the lamb1t mouse to human dystonia is tentative, but supported by a compatible distribution of gene expression in subcortical motor circuits (and not in motor neurons), and by empirical similarities that are likely to reflect shared cellular mechanisms.

There is a common premise that basal ganglia and brainstem select the muscles that co-contract in dystonia, for example by reducing the inhibition of competing pattern generators in pallidum (*Mink, 2003*), but this mouse presents evidence for a modified perspective. Although the basal ganglia may control the inhibition of unwanted activity, overactive pattern generators in the spinal cord appear to be the proximal cause of co-contraction in lamb1t. It is likely that some spinal central pattern generator circuits normally generate co-contraction, such as for postural control (*Blood, 2008*), just as others generate alternating contraction for locomotion (*Zhang et al., 2014*). Lamb1t's aberrant spinal activity may be essential for its manifestation of dystonic movements and postures. In awake lamb1t mice, we predict that the lapses in motor control that produce dystonic movements are failures (due to mutation) of supraspinal inhibitory control originating in familiar dystonia circuits, and modulated by arousal or stress.

## Materials and methods

### Origin and breeding

All animal research followed the NRC *Guide for the Care and Use of Laboratory Animals* and the policies of the Massachusetts General Hospital or Albert Einstein College of Medicine: MGH IACUC approved protocol 2011N000108, and Albert Einstein approved protocol 20130801. The lamb1t mouse arose in a colony carrying a knockout allele for an unrelated gene, *Fxyd2*. The strain (B6N.129(Cg)-Fxyd2$^{tm1Kdr}$) had been obtained from G.M. Kidder (Univ. of Western Ontario) (*Jones et al., 2005*). We had back-crossed the colony to C57BL/6NCrl (Charles River Laboratories, Wilmington, MA) for six generations (currently 12 generations) and het x het matings produced WT, het, and *Fxyd2* KO littermates. The lamb1t proband was WT for *Fxyd2*, and was the only symptomatic mouse among 29 siblings. We segregated the dystonic proband and descendants as a separate colony without the *Fxyd2* gene modification. Dominant inheritance was readily established by breeding with WT C57Bl/6NCrl mice. Every new generation was produced from colony members mated

with WTs obtained from the supplier, and because penetrance remained 100% through six generations, it is evident that the phenotype is monogenic on the C57Bl/6NCrl background. The strain name for lamb1t is proposed to be B6N-Lamb1[tr57/Swea].

Mice were housed in ventilated racks on a 12-hr light-dark cycle. Ear punches were collected under isoflurane anesthesia at weaning both to mark individuals and as a source of DNA for genotyping. Both sexes bred with a success rate equivalent to other C57Bl/6 colonies housed in the same room, in contrast with the reduced breeding success of the *Fxyd2* strain, which has pancreatic islet β cell hyperplasia and elevated insulin (*Arystarkhova et al., 2013*). Longevity was not affected by the mutation: we observed the proband and two male sibs for 2 ½ years, and others for >12 months.

## Behavioral tests

All behavioral tests were approved by the IACUC of the Massachusetts General Hospital. We performed all experiments between 2 pm and 7 pm except sleep observations, which were from 8 am to 3 pm. For gait analysis, a DigiGait apparatus (Mouse Specifics, Framingham, MA) was used for ventral plane videography of mouse gait kinematics on a moving transparent treadmill belt. Mice were tested in late afternoon close to the beginning of the active period. Each mouse was given 4–6 trials at 15 cm/s, a relatively slow treadmill speed even compared to mice with ethanol or basal ganglia toxin-induced gait impairment (*Amende et al., 2005*; *Kale et al., 2004*). The balance beam was a 120 cm rectangular rod 1.6 cm in width held 30 cm above the padded surface, resting at one end on the edge of the home cage, which was shaded with a dark box. The accelerating Rotarod test (Rotamex, Columbus Instruments, Columbus, OH) entailed accelerating the rod from 4 rpm to 40 rpm over 180 s. Mice were trained for 2 days, 2 trials per day, with a 5-min break between trials. The results of the two trials were recorded on the third day. To test swimming speed, we developed a test that was named Olympic pool. A 86 x 46 cm plastic storage box (Sterilite 1764; *Figure 3D*) was divided into four equal lanes with opaque plastic dividers, and a 5-cm-wide submerged Styrofoam platform was fastened at one end of each lane. The pool was filled with room temperature water 20 cm deep, just over the platform. The starting point was marked so that the total length for mice to swim was 71 cm (28"). Training consisted of putting the mouse in the water at the opposite end once and allowing it to find the platform. On subsequent trials they would swim to it in seconds, and the measured parameter was swimming speed. For activity analysis, we used an OPTO-Varimex Minor activity meter (Columbus Instruments) to monitor activity for 10 min by optical beam-breaks, using an empty rat cage (37 x 25 x 19 cm) as the chamber. Mice were introduced to the activity chamber as a novel environment, and beam-breaks were recorded for the first 10 min. The groups were matched for average age (WT average $130 \pm 27$ days; mutant $127 \pm 28$). WT mice were 11% heavier than lamb1t in males and 5% heavier in females. To detect impairment of forelimb function, the adhesive removal test was performed, in which a sticky square (5 mm squares of Post-it) was applied to the forehead. The time to brush it off was recorded (with a 60 s cut-off), and two trials were averaged. Replicate trials of individuals were combined and averaged. To assess grip strength and skill, a food rack from a rat cage was prepared by covering the edges with a 3-cm-wide masking tape to prevent mice from climbing to the top. Each mouse was placed in the middle and the rack was shaken laterally three times to make them grip it well. It was then inverted 30 cm above a well-padded surface. The time spent clinging to (or climbing around on) the rack was recorded, and the test was terminated after 60 s.

## Statistics

Statistical analyses were performed with GraphPad Prism software (La Jolla, CA). Genetics data were parametric and analyzed by Student's two-tailed t-test. Behavioral experiments were not designed a priori for parametric statistical analysis because, besides normal sources of variance, there was additional variance stemming from lamb1t's responses to stress and their ability to also bring motor behavior under control. The behavioral performance of the mutant was sometimes neither normally distributed nor of equal variance compared to WT, often in the tests measuring obvious abnormalities. All the available mice of appropriate age were tested since there was no ethical reason to limit group size. Statistical significance was estimated using two-way ANOVA followed by software-provided post-hoc multiple comparison tests, or two-tailed independent Student's t-test, and considered significant at $p<0.05$. Where error bars are shown, compiled data are reported as means ±

SEM. Where variance was large, the data are shown as scatter plots. No outlying data points were excluded except in two cases: when a mouse was unable to stay on the elevated beam at all, and on the rotarod, when a mouse turned around to face the wrong way. All replicates were biological replicates; when duplicate trials of individuals were averaged and used as single data points, it is stated in the legend. Because the lamb1t symptoms in the C57Bl/6N strain background are obvious, it was not possible to perform behavioral tests blinded. Although we used many of the behavioral tests on B6/FVB hybrids to assess phenotype, only C57Bl/6N strain results are presented here, except that hybrids were also included for observations of hindlimb activity during sleep. Cerebellar neuron firing rates were analyzed by the Mann-Whitney test, and data are reported as mean ± SEM.

## Genetics

For mapping the locus of the gene, we generated hybrids between mouse strains. FVB/NCrl (Charles River Laboratories) was selected because of an adequate number of different SNPs, and FVB's robust breeding ability. N1 hybrids were generated using either female or male B6 mutants paired with FVB mates. Following that, symptomatic hybrids were bred successively to FVB mice, producing N2 to N4 generations. In addition, N3 mice were intercrossed (N3F1) and backcrossed again (N3F1N1).

In the hybrids, symptoms were not as easy to detect, and so they were subjected to a battery of tests from 3 to 8 weeks of age, and scored for the presence and repeatability of symptoms. The criteria were display of symptoms upon awakening; during tail suspension; on the balance beam; during sleep; during isoflurane anesthesia; during a 30 s swim; and after vibration of the knee joint. Vibration was administered with a battery-operated fingernail polisher (Nail Wizard) with plastic tubing to cushion the tip. Cumulative scores were used to rank hybrid mice as moderately affected, weakly affected, or no detectable symptoms.

We submitted DNA from symptomatic hybrids (17 N2 and 7 N3F1N1 individuals) for SNP mapping to identify the locus. SNP mapping of recombinations was performed on the Illumina mouse medium density linkage panel of 1449 SNPs at the Centre for Applied Genomics at SickKids, University of Toronto. 833 of the SNPs on the panel differed between C57Bl/6 and FVB strains. Through six successive generations of backcross to FVB, 157 out of 302 mice on the hybrid background were *Lamb1* mutant as determined by SNP mapping or AS-PCR, indicating no in utero lethality for heterozygotes.

Exome sequencing was performed to detect variants (point mutations and small indels) that might be causative. Genomic DNA was purified with the Qiagen DNeasy Blood and Tissue Kit. Exome DNA was captured with the Agilent SureSelect Mouse All Exon system, and exome sequencing was performed on an Illumina HiSeq 2500 instrument at the Broad Institute of Harvard and MIT. Variants were called using GATK software (*DePristo et al., 2011*). Post-sequence analysis and variant calling was conducted by ContigExpress (New York, NY). Sequence surrounding the only nonsynonymous coding variant in the locus was captured by PCR from each candidate mouse separately with the following primers: F- GCAGACTCTAGATGGCGAACTT, R- TGTAGATGACTGCCTCGGTTT, and purified with the Qiagen QIAquick PCR Purification kit. Sanger sequencing was performed at the DNA Core of the Center for Computational and Integrative Biology at Massachusetts General Hospital. Note that an upstream methionine in the reference sequence NM_008482 may be incorrectly identified as the initiation methionine; we numbered the residues from the second methionine corresponding with other species.

Allele-specific PCR primers for *Lamb1* mutation at T5460A were designed with the help of Batch-Primer3 V. 1.0 (http://probes.pw.usda.gov/batchprimer3/). Allele-specific primers utilized 26 bases upstream and 20 bases downstream of the mutation (*Figure 5D*). The genotyping PCR reaction was done with ear punch or tail tip tissue, using the REDExtract-N-Amp tissue PCR kit (Sigma-Aldrich, St. Louis, MO) with half the recommended volumes. Thermal cycling was in thin-walled tubes (Molecular BioProducts, Fisher Scientific) capped with mineral oil, in an MJ Research PTC-100 thermal cycler. The conditions were 3 min 95°C, 32 cycles of 30 s 95°C denaturation, 30 s 63°C annealing, 30 s 72°C extension, and 2 min 72°C, and the products were resolved on 1.2% agarose gels. Relatively high annealing temperature was essential for allele specificity, although it reduced yield. The primer set used for routine genotyping of litters with WT and obligatory heterozygote pups were F-outside, GCCCAAGTACTTTGATATTCCTC; R-outside, TTTCACAAGTTCATCTCCACAGA, and R-mutant,

GCTTGCTGTTAGCTTGAGCCT. Reactions of (F-outside + R-outside) and (F-outside + R-mutant) were run in separate tubes.

## Protein analysis

Choroid plexus was dissected under a microscope from brains submerged in ice-cold Dulbecco's PBS. It was taken from lateral ventricle using #5 forceps after dorsal incision through the corpus callosum to access the ventricles, and from 4th ventricle by rostrally folding the cerebellum back from the brainstem. Because laminin is an ECM protein, crude homogenates of the choroid plexus, cerebellum, and sciatic nerve were used for analysis of the protein on immunoblots. The buffer was 250 mM sucrose, 20 mM Tris, 1 mM EDTA, pH 7.2 containing 1 Roche protease inhibitor tablet per 50 ml, and homogenization was with a small Tenbroeck homogenizer on ice. Protein concentration was determined by BCA assay (Pierce). Gel electrophoresis was with NuPage 4–12% polyacrylamide gradient MES gels (Life Technologies). Twenty-five microgam of protein was loaded per lane from WT or lamb1t. The proteins were transferred to nitrocellulose and stained with laminin β1-specific antibody (LTE) from NeoMarkers at a dilution of 1:500, followed by HRP-conjugated secondary antibody. Development was with Pierce West Dura luminal reagent or WesternBright, Advansta (Menlo Park, CA), and images were collected with a GE Healthcare LAS4000 imaging system.

Theoretical analysis of the impact of the mutation on the coiled-coil was done with MultiCoil, which performs computations based on a database of crystal structure data of three- and two-stranded coiled coils (Wolf et al., 1997).

## Anatomical database evidence of *Lamb1* expression in the mouse brain and spinal cord

*Lamb1* expression data in the nervous system were found in the Allen Brain Atlas, the Allen Institute for Brain Science (Lein et al., 2007) (http://mousespinal.brain-map.org/), and GENSAT (The Gene Expression Nervous System Atlas [GENSAT] Project, The Rockefeller University [New York, NY]) (Schmidt et al., 2013). (http://www.gensat.org/daily_showcase.jsp).

## Neurophysiological studies

All animal procedures were approved by the IACUCs of the Massachusetts General Hospital or Albert Einstein College of Medicine. Nerve conduction velocity was measured in mice anesthetized using a constant inhaled mixture of oxygen and isoflurane administered by a VetEquip instrument through a nose cone. Animals were placed on a heating pad to maintain their core temperature at 37°C. Hind legs were shaved with a razor and then cleaned using alcohol pads. A pair of monopolar disposable 28G needle electrodes and ring electrodes (CareFusion) were lightly coated with electrode gel (SignaGel) and used for stimulation and recording, respectively. The active recording ring electrode was placed over the gastrocnemius muscle, with the reference electrode over the tendon. The stimulating cathode was placed 5 mm proximal to the recording electrode in the midline of the posterior thigh. The anode was placed subcutaneously in the midline over the sacrum. A surface electrode (CareFusion) was grounded on the mouse's tail. We performed the studies using a portable electrodiagnostic system (Cardinal Synergy). For the motor nerve conduction studies, the low-pass filter was set at 30 Hz, and the high-pass filter was set at 10 kHz. The nerve was stimulated with single square-wave pulses of 0.1 ms duration. Supramaximal responses were gradually generated, and maximal responses were obtained with stimulus currents <20 mA (most often <10 mA). The distance between distal and proximal stimulation sites was measured with a millimeter-graduated tape measure. Data were acquired with a sensitivity of 20 mV/division and sweep speed of 3 ms/division. The distal latency, distal and proximal compound motor action potential (CMAP) amplitudes, distal and proximal CMAP durations (measured from onset of initial negative deflection to initial return to baseline), and conduction velocity were determined for each nerve studied.

A tail immersion test was used to assess the nociceptive reflex. Naive mice were held above a water bath at 51°C, sitting unrestrained on the top of the hand with the tail pointing down and held in position between two fingers. Then, 1.5–2 cm of the tail was immersed, and the latency to flick it out of the water was recorded.

Sections of sciatic nerve were fixed in fresh periodate-lysine-paraformaldehye (McLean and Nakane, 1974), washed with Dulbecco's PBS and infiltrated with 30% sucrose (30 g up to 100 ml),

cut with a cryostat, then stained with 0.2% toluidine blue and examined with a Nikon Diaphot inverted phase-contrast microscope without phase ring. Additional samples were fixed with formalin, sectioned, and stained with hematoxylin and eosin with comparable results.

Two-needle EMG was performed in anesthetized mice as described above for nerve conduction velocity. A ground self-adhesive gelled surface electrode was placed over the tail. Potentials were recorded from several sites of the hindlimb muscles with concentric needle electrodes (30G) using a gain of 50 µV/division and a band-pass filter with low and high cut-off frequency settings of 10 or 20 and 10,000 Hz, respectively. The entire recording process took 20–30 min per mouse. EMG recordings were done as previously described (*Xia et al., 2012*).

Spinal transection was performed under continuous controlled isoflurane/oxygen anesthesia as above. The top of the spine was exposed by dissection, a laminectomy was performed at L1 to L3 with sharp scissors, and the spinal cord was transected with a narrow scalpel. Motor responses under constant anesthesia were observed and filmed for up to 2 min, and euthanasia was then performed by increasing the isoflurane followed by cervical dislocation. We verified completeness of the transection by inspection after death.

For in vivo electrophysiology, mice were anesthetized with isoflurane and implanted with a custom-made L-shaped metal bracket fixed onto the skull with three bone screws (Plastics one Inc.) and dental cement (M&S Dental Supply). A recording area 2 mm wide was drilled in the skull on top of the cerebellum at AP: −6.25 mm; ML: ± 1.7 mm. The recording area was surrounded with dental cement and covered with surgifoam and bone wax (Ethicon). Mice were allowed to recover 24 hr before recording sessions. For recordings, the mouse was immobilized by fixing the head bracket with a screw attached to the stereotaxic frame. The bone wax and surgifoam were removed from the recording area. Single-unit neural activity was recorded extracellularly in awake head-restrained mice using a carbon fiber electrode (Kation Scientific, 0.4–1.2 MΩ). The electrode was advanced into the cerebellum to target the Purkinje cells and neurons of the deep cerebellar nuclei. Cell types were identified based on location and the presence of complex spikes in Purkinje cells. Signals were band-pass filtered (200 Hz-20 kHz), amplified (2000 x), and digitized (20 kHz). Waveforms were sorted offline (Plexon) using principal component analysis. During recordings the mouse was closely monitored. Abnormal postures could clearly be seen and these periods were noted. Cells recorded during these episodes were noted as 'abnormal postures', while all other cells recorded when the mouse did not show abnormal postures were categorized as 'no postures'.

## Acknowledgements

Supported by US Department of Defense grant PR100747 and US National Institutes of Health grant NS081558 to K.J.S.; NS050808 and NS079750 to K.K.; and NS058949 and NS058949S1 to A.B., K.J. S., L.J.O., and K.K.. Exome sequencing was supported by the Mouse Mutant Resequencing Project (now discontinued) of The Broad Institute. We would like to thank Jeremy Johnson of The Broad Institute for coordinating the exome sequencing and Douglas Zhang of ContigExpress for variant calling. We also thank David R. Beier of Seattle Children's Research Institute, Tara Patton of the Centre for Applied Genomics, SickKids, and Nutan Sharma of Massachusetts General Hospital for advice.

## Additional information

### Competing interests

TGH owner of the company that has commercialized the gait analysis instrumentation described AB performs research at Wake Forest with grants from Allergan, Ipsen, and Merz, and has consulting relationships with Allergan and Concerta. Her conflict of interest is managed by Wake Forest School of Medicine The other authors declare that no competing interests exist.

## Funding

| Funder | Grant reference number | Author |
|--------|------------------------|--------|
| National Institute of Neurological Disorders and Stroke | NS058949 | Allison Brashear<br>Laurie J Ozelius<br>Kathleen J Sweadner |
| National Institute of Neurological Disorders and Stroke | NS058949S1 | Allison Brashear<br>Laurie J Ozelius<br>Kamran Khodakhah<br>Kathleen J Sweadner |
| National Institute of Neurological Disorders and Stroke | NS050808 | Kamran Khodakhah |
| National Institute of Neurological Disorders and Stroke | NS079750 | Kamran Khodakhah |
| Congressionally Directed Medical Research Programs | PR100747 | Kathleen J Sweadner |
| National Institute of Neurological Disorders and Stroke | NS081558 | Kathleen J Sweadner |

The funders had no role in study design, data collection and interpretation, or the decision to submit the work for publication.

## Author contributions

YBL, AT, JS, KJS, Conception and design, Acquisition of data, Analysis and interpretation of data, Drafting or revising the article; EA, KK, Conception and design, Analysis and interpretation of data; TGH, Acquisition of data, Analysis and interpretation of data, Drafting or revising the article; AB, LJO, Conception and design, Analysis and interpretation of data, Drafting or revising the article

## Author ORCIDs

Kamran Khodakhah, http://orcid.org/0000-0001-7905-5335
Kathleen J Sweadner, http://orcid.org/0000-0002-2817-5262

## Ethics

Animal experimentation: All animal research followed the NRC Guide for the Care and Use of Laboratory Animals and the policies of the Massachusetts General Hospital or Albert Einstein College of Medicine: MGH IACUC approved protocol 2011N000108, and Albert Einstein approved protocol 20130801.

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
