## [Decision Letter]

Thank you for submitting your work entitled "A dystonia-like movement disorder with brain and spinal defects caused by mutation of the mouse laminin β1 subunit *Lamb1*" for consideration by *eLife*. Your article has been reviewed by 3 peer reviewers, and the evaluation has been overseen by a Reviewing Editor and a Senior Editor. Our decision has been reached after consultation between the reviewers. Based on these discussions and the individual reviews below, we are unable to publish your work in its current form but we are open to publishing the paper if you address the key points raised by the reviewers. We realize that the issues raised should be straightforward to address but they are likely to take more than the two months that we usually allow for revisions.

Summary:

The authors propose a new mutant mouse model of dystonia-like symptoms. Mapping and exome sequencing revealed a mutation in the *Lamb1* gene. They provide extensive behavioral phenotyping of the mice. Electrophysiological experiments detected irregular firing from a circuit of 2 *Lamb1*-expressing neurons, Purkinje cells and their deep cerebellar nucleus targets.

Essential revisions:

This mouse seems to be a good model for dystonia-like symptoms. A fair amount is known about *lamb1* including it's having a role in synapse structure and plasticity. The reviewers appreciate the important work included in this manuscript but felt that additional electrophysiological data or molecular data would significantly strengthen the paper. In the spirit of not asking for extensive additional experiments, discussion between reviewers led to a consensus that one of the following would be sufficient:

1) Physiology: Recording of activity of DCN and Purkinje neurons while the animals were experiencing dystonic symptoms and to correlate the physiological findings with the behavior. This could provide some cellular insight in the described dysfunction.

2) Molecular: Providing direct evidence that any of observed features are related to impaired integrin signaling/adhesion to integrin receptors.

*Reviewer #1:* The authors describe a new mutant mouse, *lamb1*, which has multiple motor abnormalities, many if not most of which are typically associated with dystonia. The model has many parallels to the human disorder of dystonia, with childhood onset, initially progressive then later stable symptoms, stress and action-induced symptoms, and gait abnormalities. The manuscript's strengths include the large number of behavioral assays included, the relatively large number of animals used in these assays, controls for peripheral nervous system involvement, and the inclusion of very helpful videos.

1) The article is written clearly, but in clinical language that might be harder for a non-clinician basic scientist to understand or appreciate. I think the gestalt of the phenotype (multiple motor manifestations of involuntary movement) is more important to convey than the nitty-gritty details of the clinical phenotype. Though it is important to mention the lack of certain features (neuropathy, myotonia), these terms could alienate the non-clinician and prevent them from understanding the gestalt.

2) While the relatively conservative interpretations provided by the authors are very reasonable, in many places the results could be framed in a larger context (even without overinterpretation of the data) and thus make the findings more interesting and thought-provoking to other scientists in the field, especially to those trying to understand the normal function of motor circuitry (such as basal ganglia, cerebellum, and brainstem/spinal cord).

3) The authors could convey the importance of this work better by highlighting the lack of phenotypically penetrant mouse models of dystonia. This model would provide one of the very few available to test circuit and cellular/molecular hypotheses regarding the mechanism of dystonia generation. The Introduction would benefit from putting their discoveries in this context. It also would be useful to put some of the material currently in the Discussion into the Introduction instead (i.e. some of the basic information regarding laminin and its role in synapses) so as to give the basic scientist a larger framework for interpreting their data.

4) The beam test in Figure 3 was presented very well, showing the development of the phenotype with age. However, the authors do not mention or include in the figure whether the phenotype is present even earlier, as in 2 weeks of age. If this data was collected, it would be very interesting to include.

5) I found the reproduction of expression data from the Allen Brain Atlas odd. I think this previous work could either be cited in the text or the authors could reproduce the key findings themselves in their model. The authors could instead schematize the high-expression regions if they think it is important to show this visually. Also, the authors make vague statements about the striatal expression of *lamb1* – in what interneuron population do they believe it is expressed? If they have a hypothesis, they should test this directly by colabeling for the standard interneuron markers (NPY, parvalbumin, SST, calretinin, choline acetyltransferase).

*Reviewer #2:* This study describes a mouse mutant with truncated beta1 laminin. Mice exhibit intermittent dystonic hindlimb movements and postures. The authors ruled out other diagnoses such as myotonia, neuromyotonia, neuropathy, muscular dystrophy, and spasticity. Recordings in awake mice revealed irregular firing from deep cerebellar nucleus neurons.

Although this study is very interesting from phenomenological point of view and clinical relevance, and offers a stimulating discussion of possible molecular and cellular mechanisms underlying these phenomena, it does not address any mechanisms. The impact of study could be significantly increased (in a short time) by providing direct evidence that any of observed features are related to impaired integrin signaling/adhesion to integrin receptors and/or by demonstration of altered synaptic plasticity at synapses between Purkinje cells and their deep cerebellar nucleus targets.

*Reviewer #3:* The authors developed and proposed a new *Lamb1* mouse for dystonia-like movement disorder model. Motor behavior tests were well performed and showed dystonic behaviors in *Lamb1* mice; however, mutant mice do not demonstrate some typical abnormal myotonia. The conclusions drawn from the results are a bit unclear. A comparative immunohistochemistry of *Lamb1* between WT and mutant mice is highly recommended, not just reproduced. Without IHC results of mutant mice, value of this paper will be largely diminished. The manuscript should also be reconstructed to avoid mix-up results, figure legends and discussions.

---

## [Author Response]

For essential revisions, we addressed point 1, Physiology. We now present electrophysiological recordings of the mutant mice while they were and were not experiencing dystonic symptoms. This showed that the abnormal firing occurred during the symptoms and not in their absence. As a result, the corresponding figure was completely restructured, and we have added a table so that the reader can more readily compare and evaluate the expanded data and statistics. The figure shows box and whisker plots with medians, as suggested by a reviewer, while the table presents the means. Both are useful for critical evaluation.

Reviewer #1:

*1) The article is written clearly, but in clinical language that might be harder for a non-clinician basic scientist to understand or appreciate. I think the gestalt of the phenotype (multiple motor manifestations of involuntary movement) is more important to convey than the nitty-gritty details of the clinical phenotype. Though it is important to mention the lack of certain features (neuropathy, myotonia), these terms could alienate the non-clinician and prevent them from understanding the gestalt.*

We have added a few words to include all readers when clinical terminology is relevant.

*2) While the relatively conservative interpretations provided by the authors are very reasonable, in many places the results could be framed in a larger context (even without overinterpretation of the data) and thus make the findings more interesting and thought-provoking to other scientists in the field, especially to those trying to understand the normal function of motor circuitry (such as basal ganglia, cerebellum, and brainstem/spinal cord).*

We have added some text to broaden the context.

*3) The authors could convey the importance of this work better by highlighting the lack of phenotypically penetrant mouse models of dystonia. This model would provide one of the very few available to test circuit and cellular/molecular hypotheses regarding the mechanism of dystonia generation. The Introduction would benefit from putting their discoveries in this context. It also would be useful to put some of the material currently in the Discussion into the Introduction instead (i.e. some of the basic information regarding laminin and its role in synapses) so as to give the basic scientist a larger framework for interpreting their data.*

We were criticized for discussing the lack of symptoms in genetically engineered dystonia mice in the review of an earlier incarnation of this study, and accused of “bashing” other’s work. We have reintroduced the subject lightly, citing some recent work. We now included some laminin background in the Introduction, as suggested.

*4) The beam test in Figure 3 was presented very well, showing the development of the phenotype with age. However, the authors do not mention or include in the figure whether the phenotype is present even earlier, as in 2 weeks of age. If this data was collected, it would be very interesting to include.*

We have clarified that symptoms were not detectable before P17-P28.

*5) I found the reproduction of expression data from the Allen Brain Atlas odd. I think this previous work could either be cited in the text or the authors could reproduce the key findings themselves in their model. The authors could instead schematize the high-expression regions if they think it is important to show this visually. Also, the authors make vague statements about the striatal expression of lamb1 – in what interneuron population do they believe it is expressed? If they have a hypothesis, they should test this directly by colabeling for the standard interneuron markers (NPY, parvalbumin, SST, calretinin, choline acetyltransferase).*

We included the Allen Brain Atlas and GENSAT data because some readers would not find the original data easy to find and all would have to devote time to it. Also ABA data sometimes disappears, to be replaced by new samples, and some of those are not as good. We think there is high value to the reader in having one page of relevant and labeled images in the paper. Identification of the striatal subpopulation, and a subpopulation in the dorsal horn, is planned and awaiting funding.

Reviewer #2:

*Although this study is very interesting from phenomenological point of view and clinical relevance, and offers a stimulating discussion of possible molecular and cellular mechanisms underlying these phenomena, it does not address any mechanisms. The impact of study could be significantly increased (in a short time) by providing direct evidence that any of observed features are related to impaired integrin signaling/adhesion to integrin receptors and/or by demonstration of altered synaptic plasticity at synapses between Purkinje cells and their deep cerebellar nucleus targets.*

We agree absolutely that these are the right directions, however the paper is already packed with novel findings and a variety of methodologies. We hope that the addition of new data is satisfactory. Thank you for understanding.

Reviewer #3:

*The authors developed and proposed a new Lamb1 mouse for dystonia-like movement disorder model. Motor behavior tests were well performed and showed dystonic behaviors in Lamb1 mice; however, mutant mice do not demonstrate some typical abnormal myotonia. The conclusions drawn from the results are a bit unclear. A comparative immunohistochemistry of Lamb1 between WT and mutant mice is highly recommended, not just reproduced. Without IHC results of mutant mice, value of this paper will be largely diminished. The manuscript should also be reconstructed to avoid mix-up results, figure legends and discussions.*

IHC would be nice, but the reviewer should not expect to see differences between mutant and wild type, because the truncation is very small and the truncated protein is demonstrated to be expressed. Laminin IHC is also notoriously hard to obtain for two reasons: the native protein is not very immunogenic because the exposed structures are extracellular, and staining is diffuse, again because laminin is secreted and deployed to the matrix between cells. The published work (L Gudas) used a beta-galactosidase-expressing mouse that does not exist anymore. What we plan going forward is to utilize genetically-modified mice to label the neurons for experiments to identify subclasses of neurons that express laminin β1.